# Train 'n Trade: Foundations of Parameter Markets

**Tzu-Heng Huang, Harit Vishwakarma, Frederic Sala**
Department of Computer Science
University of Wisconsin-Madison
{thuang273, hvishwakarma}@wisc.edu, fredsala@cs.wisc.edu

## Abstract

Organizations typically train large models individually. This is costly and time-consuming, particularly for large-scale foundation models. Such vertical production is known to be suboptimal. Inspired by this economic insight, we ask whether it is possible to leverage others' expertise by trading the constituent parts in models, i.e., sets of weights, as if they were market commodities. While recent advances in aligning and interpolating models suggest that doing so may be possible, a number of fundamental questions must be answered to create viable parameter markets. In this work, we address these basic questions, propose a framework containing the infrastructure necessary for market operations to take place, study strategies for exchanging parameters, and offer means for agents to monetize parameters. Excitingly, compared to agents who train siloed models from scratch, we show that it is possible to mutually gain by using the market, even in competitive settings. This suggests that the notion of parameter markets may be a useful paradigm for improving large-scale model training in the future.

## 1 Introduction

Costs to build powerful state-of-the-art machine learning models, such as foundation models (e.g., GPT-3 [1], T5 [2], PaLM [3], BLOOM-176B [4] and OPT-175B [5]), have increased enormously. These costs can easily reach millions of dollars and even exceed that amount. For example, training the 11 billion-parameter T5 model is estimated to take around 1.3 million dollars for a single training run [6]. Unfortunately, few organizations and fewer individuals are sufficiently well-capitalized to afford such training costs.

One approach to reduce expense is to broadly distribute training workloads, such as in decentralized training [7, 8, 9, 10]. However, this is limiting; even in the decentralized setting, participants must first agree to train a shared model and at least minimally coordinate the training process. For this reason, such techniques cannot be applied when organizations develop different models for different purposes on different timelines. In these scenarios—the most common solution in large-scale model development—models are trained individually regardless of high cost.

A natural question is whether such *vertical production* can be broken down into parts that can be more easily built, re-used, and exchanged. Taking inspiration from other areas of manufacturing, we observe that most products are not built in vertical silos, but from components that are traded in markets. Economic agents, even when competing against each other, buy, sell, and trade such components to leverage the expertise of other agents so that production costs can be lowered.

This leads us to ask whether subsets of trained weights can be thought of as constituent parts to be bought and sold on *parameter markets*. Such markets may provide mutual benefits for both buyers and sellers. Buyers are able to purchase well-trained parameter sets directly as commodities to leverage the training expertise of others and then use them to improve model performance. Sellers (i.e., owners of partially or fully-trained models) are able to monetize parameters as a second profit center, in addition to the downstream activity enabled by using their models.

37th Conference on Neural Information Processing Systems (NeurIPS 2023).

**Challenges and Proposed Approach.** How can we build and use such markets? To answer this, we must first overcome several obstacles. An immediate challenge is the notion of *alignment*: models trained in isolation may not have parameter sets that correspond in any natural way, especially if these models have differing purposes. Excitingly, recent work suggests that it is possible to align model components and then merge them via linear interpolation [11, 12, 13, 14, 15, 16, 17]. Similarly, it is known that training data can be potentially recovered from weights or gradients so privacy is an additional challenge [18, 19, 20]. We tackle the orthogonal question:

> If model alignment and sufficient privacy can indeed be assured, how can a viable parameter marketplace be designed?

There are three fundamental challenges in doing so:

1. *How should agents (users in the market) decide to perform transactions?* Discovering and verifying useful model parameter sets on the market without prior knowledge is challenging for buyers. To address this issue, we introduce a trusted third-party organization, the *broker*. The broker enables a "try-before-purchase" mechanism for buyers to examine the quality of parameters. This is a common approach in the existing works on *data* marketplaces [21, 22] as it allows buyers to evaluate the quality of data before purchase. Doing so with parameters rather than data presents additional complications that must be resolved.

2. *What rewards may agents gain?* An important consideration when using such a framework is whether *agents can expect to see any gains*. Indeed, if the process of exchanging and validating parameter sets does not yield any improvements when compared to siloed training, there is no reason to participate in parameter markets. We provide theoretical and empirical analyses validating the improvements gained from participating in such markets.

3. *How to monetize model parameters in a competitive market?* In settings where parameters are bought and sold, rather than bartered, it can be challenging to price these assets. Both the seller and the buyer may not have a clear understanding of the other's valuation of the parameters, which makes it difficult for each to maximize their revenues in a trade. To address this issue, we apply a Bayesian-optimal pricing mechanism [23] to provide valuation of parameter sets and study Nash bargaining [24] to find market prices in negotiation.

**Results and Contributions.** We propose and formulate a viable marketplace to trade parameters for machine learning models. We validate it in both theoretical and empirical settings. Theoretically, in basic scenarios we show how agent training converges faster through purchasing parameters in the market. We offer bounds on the improvement gained via trading when training linear models. Empirically, we conduct experiments in a variety of practical scenarios to validate the framework's effectiveness. We demonstrate that compared to agents who stay outside the market and train models in isolation, participating in parameter trading, even trading subsets of the full set of model parameters, provides benefits to efficient model training and better model performance. For example, when training and trading parameters of ResNet20 on TinyImageNet, two agents improve their performance by gaining accuracy improvements of **+10.02%** and **+15.93%** versus separate training. We also demonstrate the success of price estimation to monetize parameters and negotiate prices.

## 2 Related Works

First, we describe two related concepts: *data* and *model*—as opposed to *parameter*—marketplaces. We then give background on model alignment techniques, which are used in our framework.

**Data Marketplaces.** Data is a key ingredient in machine learning pipelines. There is a rich vein of work proposing infrastructure to trade data as a commodity [21, 22, 25, 26, 27, 28, 29]. Such marketplaces have also been instantiated in industry, including in services such as Amazon Web Services (AWS) Data Exchange, Microsoft's Azure Data Marketplace, and Google's Cloud Data Catalog. Such research; however, cannot be directly applied to trading parameters. It is relatively easy to evaluate the valuation of a data trade using basic statistical measurements. In contrast, the value of parameters is challenging to measure, as it can only be determined after testing and depends on the model's performance.

**Model Marketplaces.** There have been several efforts to build markets to trade *entire model instances* trained by a centralized broker [30, 31]. Two major obstacles that need to be overcome are determining the value and pricing models, and safeguarding privacy. To address the former issue, a noise-injection mechanism has been suggested, which involves assessing the accuracy of an optimal model with random Gaussian noise to determine its worth and creating a price-error curve for selling it on the market [30]. The latter issue has been tackled by proposing a system that apply differential privacy while still maximizing revenue [31]. In contrast to trading entire model instances for downstream use, parameter markets are far more refined, enabling each user in the market to train their own models for their own purposes while gaining from others' training runs.

**Model Alignment.** Models that are trained with different batch orders or initialization weights may not be properly aligned. Directly merging purchased parameters through interpolation may fail. The process of aligning parameters in model training is therefore critical. Recent studies have explored the geometric relationship between models [11, 12, 13, 14, 15] and proposed methods for aligning two sets of neural network parameters [16, 17]. We use such techniques as a building block in our proposed market infrastructure. Through proper model alignments, we expect agents are able to find and purchase desired parameter sets in the market.

## 3 A Framework for Parameter Markets

We provide a general description of the proposed marketplace and then discuss each component in depth.

### 3.1 General Marketplace Framework

Figure 1 depicts a two-agent version of the marketplace. Multiple agents training models for potentially different tasks seek to buy or sell sets of parameters. Buying well-trained parameter sets enables reaching a goal performance faster while selling such parameters produces profits.

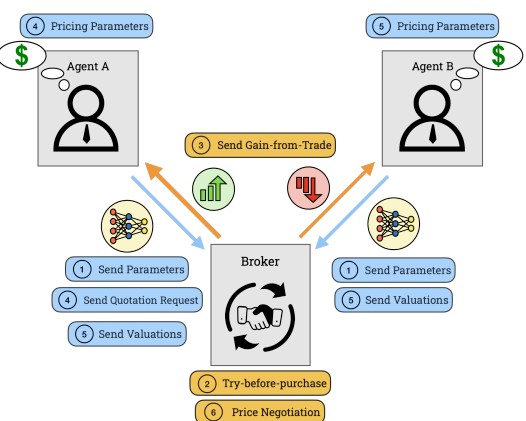

A high-level description of the trading process follows. First, agents send their own parameters to the market ①. A third party (the broker) operates a "try-before-purchase" routine to align and merge parameters for buyers ②. The broker privately informs agents of the gains or losses resulting from a potential trade using validation data ③. Based on this information, a buyer values a seller's parameters, and then makes a trading decision. If the buyer is willing to purchase, a quote request is sent out, and both sides generate and report their valuations to the broker ④, ⑤. The broker helps both parties negotiate the price until they reach an agreement ⑥. Afterwards, the broker ships parameters to the buyer and transfers the payment to the seller, completing the trade.

Figure 1: Overall workflow in a two-agent market. Blue and orange blocks represent actions taken by agents and the broker, respectively. In this example, agent $A$ is informed of a potential gain through purchasing agent $B$'s parameters. Hence, agent $A$ sends a quotation request to inquire about purchasing parameters. Then, broker helps both sides negotiate on the price of agent $B$'s parameters.

### 3.2 Market Setup

In the following sections, we discuss each foundational concept in parameter markets. We first fix some notation. For agent $u$, let $D_u = \{s_{u,i}\}_{i=1}^{n_u}$ be the samples drawn from a distribution $\mathcal{D}^u$ supported on $\mathcal{S}$. At round $t$, let $\theta_u^t \in \mathbb{R}^d$ be trained parameters that agent $u$ has access to, and let $\hat{\mathcal{L}}_u(\theta_u^t)$ be the empirical loss of agent $u$ for the corresponding model measured on data $D_u$. The

empirical loss is defined with the agents' loss function $\ell$ in a standard fashion,

$$\hat{\mathcal{L}}_u(\theta_u^t) := \frac{1}{n_u} \sum_{i=1}^{n_u} \ell(\theta_u^t, s_{u,i}).$$

While our framework can handle general settings (i.e., any well-trained parameter sets can be traded), we focus on supervised learning for ease of exposition. We assume $\mathcal{S} = \mathcal{X} \times \mathcal{Y}$ where $\mathcal{X}$ and $\mathcal{Y}$ are the instance and label spaces respectively. Thus, $s_{u,i} = (x_{u,i}, y_{u,i})$, and also denote $D_u$ as $(X_u, Y_u)$ where $X_u$ is the set of points $x_{u,i}$ and $Y_u$ is the set of corresponding labels. We drop the superscript $u$ when it is otherwise clear from the context.

**Agents and Broker.** For simplicity, suppose there are two agents $A$, $B$ in the market. Agent $A$ has $n_a$ data points $(X_a, Y_a)$, and agent $B$ has $n_b$ data points $(X_b, Y_b)$. There is also a *trusted* third party, the broker $Z$, helping the two agents operate in the market impartially. The broker has access to a validation dataset $(X_z, Y_z)$ of size $n_z$ with noiseless samples from both distributions.

In the proposed trading framework, the two agents train their models locally first and then trade with each other. At the beginning of round $t$, agents perform a standard gradient descent update using their own datasets. That is, $\dot{\theta}_u^t := \theta_u^{t-1} - \eta \nabla \hat{\mathcal{L}}_u(\theta_u^{t-1})$, $u \in \{a, b\}$, where $\eta$ indicates the step size.

**Using Purchased Parameters.** In the market, parameters can be bought or sold. To use acquired parameters, several model alignment techniques can be employed [16, 17]. For a potential trade, broker aligns seller's parameter sets to match a buyer's. Post-alignment, the parameters can be merged as a straightforward convex combination. For instance, if agent $A$ is willing to buy parameters from agent $B$, and a trade occurs eventually, the post-alignment combination will have the form $\bar{\theta}_a^t := (1 - \alpha)\dot{\theta}_a^t + \alpha \dot{\theta}_b^t$, $\alpha \in (0, 1]$. Here $\alpha$ and (for agent $B$, $\beta$) are weights indicating the portion of trained parameters from the seller side in the merged parameters.

### 3.3 Try-Before-Purchase Mechanism

However, before making any trading decision, agents are unaware of the potential benefits they can achieve through purchasing. Our proposal is for a neutral broker to implement a "try-before-purchase" mechanism which assists agents. It does so by helping them align and merge parameters then pre-evaluate their quality by using the broker's dataset $(X_z, Y_z)$. In the try-before-purchase mechanism, the broker can pick the optimized weights $\alpha$ or $\beta$ for buyer agents by minimizing the empirical loss. We write $\alpha := \arg\min_{\nu \in (0,1]} \hat{\mathcal{L}}_z((1 - \nu)\dot{\theta}_a^t + \nu \dot{\theta}_b^t)$ and $\beta := \arg\min_{\nu \in (0,1]} \hat{\mathcal{L}}_z((1 - \nu)\dot{\theta}_b^t + \nu \dot{\theta}_a^t)$.

Using the optimized purchased weights, the broker calculates and communicates to agents their gain or loss from the trade in a confidential manner. We denote the *gain-from-trade* for agent $u$ by $\Delta_u^t$, $u \in \{a, b\}$, which serves as prior knowledge to help agents make informed decisions about whether to make purchases or not. Generally, the notion of gain-from-trade is to compare relative improvement versus not trading. The trading benefits can be expressed in various ways, such as the difference between agent's loss before and after the trade. For example, it might take the form $\Delta_u^t = \hat{\mathcal{L}}_z(\dot{\theta}_u^t) - \hat{\mathcal{L}}_z(\bar{\theta}_u^t)$. Other ways to define the gain-from-trade include using the relative improvement ratio on the empirical loss or improvement ratio on the (estimated) parameter recovery error.

If the gain-from-trade $\Delta_u^t$ does not indicate any benefit for buyer agent $u$, the agent will not be willing to make a purchase. Consequently, no quote request will be sent out, and no trade will take place. In such a scenario, the parameters $\dot{\theta}_u^t$ will remain with agent $u$ until the next round of gradient descent. The final parameters at the end of round $t$, denoted by $\theta_u^t$, can be either $\bar{\theta}_u^t$ or $\dot{\theta}_u^t$. To indicate a trade, we use the indicator variable $I_u^t$:

$$\theta_u^t := (1 - I_u^t) \cdot \dot{\theta}_u^t + I_u^t \cdot \bar{\theta}_u^t \quad \text{where } I_u^t = \begin{cases} 1 & \text{if agent } u \text{ buys parameters,} \\ 0 & \text{if agent } u \text{ does not make a purchase.} \end{cases}$$

## 3.4 Valuation and Pricing Mechanism

Once the gain-from-trade has been determined and trading decisions have been made, quote requests with purchased weights are circulated throughout the market. Both buyers and sellers begin to assess various parameters in order to negotiate prices. Each agent has their own private valuation function, denoted by $v_u : \mathbb{R}^d \mapsto \mathbb{R}^+$, which quantifies their trading benefits produced by specific parameters $\theta \in \mathbb{R}^d$. The valuation function of each agent is private and is not exposed to others. For instance, agent $A$'s valuation function is denoted by $v_a(\dot{\theta}_b^t)$ for the value of agent $B$'s parameters for purchasing, and $v_a(\dot{\theta}_a^t)$ is the value of their own parameters for selling. In order to generate revenue from a trade, $v_a(\dot{\theta}_b^t)$ is the highest price that agent $A$ is willing to bid for purchase, while $v_a(\dot{\theta}_a^t)$ is the lowest price that agent $A$ is willing to ask for in order to sell.

Once valuations are completed, the broker—in their role as an impartial middleman—assists in bargaining the market price to maximize revenue for both sides. The negotiation process for prices continues until a mutually acceptable market price for parameters is reached. When a buyer offers to pay a lower price for certain parameters than the seller is willing to accept, the trade cannot be fulfilled. To analyze this negotiation process, we treat it as a Nash bargaining problem [24]. We assume that the broker is knowledgeable about the valuations of both parties (not the valuation functions themselves) and sets the price by maximizing the revenue of both agents impartially. An agent's revenue, defined as $U_u$, is derived from two sources: the profit earned from selling self-parameters and the profit gained from buying parameters. We employ the popular Cobb-Douglas function to maximize both agents' revenue [32]. We denote the market price for agent $u$'s parameters at round $t$ as $P_u^t \in R^+$, then formulate the problem accordingly. Set $U_a(P_a^t, P_b^t) := \left(P_a^t - v_a(\dot{\theta}_a^t)\right) + \left(v_a(\dot{\theta}_b^t) - P_b^t\right)$ and $U_b(P_b^t, P_a^t) := \left(P_b^t - v_b(\dot{\theta}_b^t)\right) + \left(v_b(\dot{\theta}_a^t) - P_a^t\right)$. Then we have the problem

$$\underset{P_a^t, P_b^t}{\operatorname{argmax}} \quad U_a(P_a^t, P_b^t) \times U_b(P_b^t, P_a^t)$$

$$\text{s.t.} \qquad P_a^t \in \left[v_a(\dot{\theta}_a^t), v_b(\dot{\theta}_a^t)\right], \quad P_b^t \in \left[v_b(\dot{\theta}_b^t), v_a(\dot{\theta}_b^t)\right].$$

By solving this problem, the broker can determine the difference in price, denoted by $\Delta P_{ab}^t$, using

$$\Delta P_{ab}^t = P_a^t - P_b^t = \frac{1}{2}\left(v_b(\dot{\theta}_a^t) + v_a(\dot{\theta}_a^t) - v_a(\dot{\theta}_b^t) - v_b(\dot{\theta}_b^t)\right). \tag{1}$$

The steps for solving this problem are shown in the Appendix A. The resulting price difference $\Delta P_{ab}^t$ represents the amount of money that needs to be transferred between parties.

## 4 Instantiating the Market: Concrete Examples

Now we give a concrete instantiation of the market that we described in the previous section.

**Valuations.** There are various ways for agents to define valuation functions (i.e. based on agent's preference, training budget, or model performance). Here we assume that in the market, agents to purchase use gain-from-trade $\Delta_u^t$, which can be seen as a notion of relative performance improvement, to assess the value of parameters so that $v_u(\dot{\theta}_{u'}^t) = \Delta_u^t$, where $u'$ represents their seller agent.

However, assessing the value of self-parameters for agent $u$, who is a seller, is a difficult task as there is no clear information available from the broker regarding the quality of such parameters. The best approach for a seller to maximize profit is to set a price as close as possible to the buyer's valuation, which is the highest price that the buyer is willing to pay. To arrive at this *virtual valuation*, we use the Bayesian-optimal pricing mechanism described in Myerson's work [23]. This mechanism enables the seller to monetize self-parameters. Under this Bayesian mechanism, we assume that the seller is also aware that the buyer's valuation arises from gain-from-trade, and that their valuation function is derived from a known probability distribution. We discuss these common priors in Appendix G.

Suppose the buyer's valuation has a cumulative distribution function $F_v$. If the seller sets a price of $P$, the probability that the buyer is willing to purchase is $\mathbb{P}(P < v) = 1 - \mathbb{P}(v \leq P) = 1 - F_v(P)$. The expected revenue for the seller is $P \times (1 - F_v(P))$. Hence, the optimal price to ask for can be

---

**Algorithm 1** Single Round of Parameter Trading

---

**Input:** $(X_a, Y_a), (X_b, Y_b), (X_z, Y_z), \theta^*, \theta_a^{t-1}, \theta_b^{t-1}$
**Output:** $\theta_b^t$

$\dot\theta_u^t \leftarrow \theta_u^{t-1} - \eta\nabla\hat{\mathcal{L}}_u(\theta_u^{t-1}), u \in \{a, b\}$                       ▷ agents' local training

$\bar\theta_a^t = (1-\alpha)\dot\theta_a^t + \alpha\dot\theta_b^t, \quad \alpha = \arg\min_{\nu\in(0,1]}\hat{\mathcal{L}}_z\big((1-\nu)\dot\theta_a^t + \nu\dot\theta_b^t\big)$    ▷ broker's try-before-purchase

$\bar\theta_b^t = (1-\beta)\dot\theta_b^t + \beta\dot\theta_a^t, \quad \beta = \arg\min_{\nu\in(0,1]}\hat{\mathcal{L}}_z\big((1-\nu)\dot\theta_b^t + \nu\dot\theta_a^t\big)$    ▷ broker's try-before-purchase

$\Delta_u^t = \frac{\|\dot\theta_u^t - \theta^*\|_2^2}{\|\bar\theta_u^t - \theta^*\|_2^2}, u \in \{a, b\}$                           ▷ inform agents about gain-from-trade

**if** $\Delta_b^t > 1$ **then**                                   ▷ agent $B$ sends a quotation request with $\beta$
    agent $A, B$ provide valuations to broker        ▷ agent $A$'s valuation is estimated by the bounds of $\Delta_b^t$
    **if** $v_b(\dot\theta_a^t) \geq v_a(\dot\theta_a^t)$ **then**
        transferred payment for buying $\dot\theta_a^t$ after negotiation is set to $\big(v_b(\dot\theta_a^t) + v_a(\dot\theta_a^t)\big)/2$
        **return** $\bar\theta_b^t$                                  ▷ ship merged parameters
    **else**
        **return** $\dot\theta_b^t$                                    ▷ negotiation fails
**else**
    **return** $\dot\theta_b^t$                                   ▷ agent $B$ decides not to buy

---

found by maximizing the expected revenue, and it satisfies

$$P^* := \frac{1 - F_v(P^*)}{F_v'(P^*)}. \tag{2}$$

**Linear Model Case Study.** To further illustrate a concrete example of the seller's virtual valuation, we consider pricing parameters for training linear models. For simplicity, we assume that the broker knows the true parameters $\theta^*$—though this is not necessary in practice, as these can be approximated using the broker's validation dataset. Both agents' data is sampled from the same distribution so that true parameters are identical. Additionally, we take the gain-from-trade revealed by broker to be the ratio of squared parameter estimation error. We write

$$\Delta_u^t := \frac{\|\dot\theta_u^t - \theta^*\|_2^2}{\|\bar\theta_u^t - \theta^*\|_2^2}. \tag{3}$$

If $\Delta_u^t > 1$, the agent's model is able to get closer to the true parameter ($\theta^*$) by purchasing. We use agent $A$ to illustrate the sale of their parameters $\dot\theta_a^t$ to agent $B$. First, we show how agent $A$ determines bounds for the buyer's valuation $v_b(\dot\theta_a^t)$. Recall that once agent $B$ expresses a willingness to purchase, a quote request with their purchased weight $\beta$ will be communicated to agent $A$ through the broker. Therefore, when agent $A$ evaluates self-parameters, the broker provides three pieces of information: the buyer's purchased weight $\beta$, agent $A$'s purchased weight $\alpha$, and the gain-from-trade $\Delta_a^t$. In this setting, we have that

**Theorem 4.1.** ***Bounds on Buyer's Gain-from-Trade***: *In the linear model setting, by knowing $\Delta_a^t$ and weights $\alpha, \beta$, agent $A$ can obtain bounds on the gain-from-trade of agent $B$ given by:*

$$\left(\frac{1 - \sqrt{\Delta_a^t}(1-\alpha)}{(1-\beta) + \sqrt{\Delta_a^t}(1-\alpha-\beta+2\alpha\beta)}\right)^2 \leq \Delta_b^t \leq \left(\frac{1 + \sqrt{\Delta_a^t}(1-\alpha)}{(1-\beta) - \sqrt{\Delta_a^t}(1-\alpha-\beta+2\alpha\beta)}\right)^2.$$

**Discussion.** Theorem 4.1 states that by knowing information disclosed by the broker (including purchased weights $\alpha, \beta$), the seller agent $A$ can find the upper and lower bounds of the buyer's gain-from-trade, $\Delta_b^t$. This information can then be used to estimate the value that the buyer places on the item. Using the Bayesian optimal-pricing mechanism, and taking into account the known probability distribution, the seller can estimate the price to determine their own virtual valuation.

Furthermore, the optimal scenario occurs when the seller agent values self-parameters exactly as the buyer does. In this case, based on Eq. (1), the broker will set the transfer payment as $\Delta P_{ab}^t = P_a^t - P_b^t = v_b(\dot\theta_a^t) - v_a(\dot\theta_b^t)$. Hence, the transfer payment is equal to the difference between the gain-from-trade of the two agents, where $\Delta P_{ab}^t = \Delta_a^t - \Delta_b^t$.

The proof for Theorem 4.1 is in the Appendix C. We summarize trading steps in a single round for this instantiation above (Algorithm 1), where two agents are training and trading parameters for the linear model setting. Here, agents $A, B$ act as a seller and a buyer, respectively.

# 5 Convergence Analysis

Next, we study a theoretical analysis for the *effectiveness of buying parameters*. In particular, we are interested in understanding whether participating in the market leads to faster training convergence (in the worst-case scenario). We show this holds in a simplified setting where agent $A$ always leads and never purchases parameters, while agent $B$ always purchases from agent $A$. This asymmetry could be due to various reasons including a lack of training resources for agent $B$. Here we study a setting for general L-smooth functions, which is more practical, as the broker doesn't need to be knowledgeable about the true parameter $\theta^*$. We assume the broker's loss is lower than agents' losses, in particular, $\hat{\mathcal{L}}_z(\theta) \leq \hat{\mathcal{L}}_a(\theta)$ and $\hat{\mathcal{L}}_z(\theta) \leq \hat{\mathcal{L}}_b(\theta), \forall \theta \in \mathbb{R}^d$. We take the gain-from-trade $\Delta_u^t$ by using the subtraction of empirical loss before and after a trade. We write it as

$$\Delta_u^t = \hat{\mathcal{L}}_z(\dot{\theta}_u^t) - \hat{\mathcal{L}}_z(\bar{\theta}_u^t), u \in \{a, b\}. \tag{4}$$

**Theorem 5.1.** *For all agents $u \in \{a, b, z\}$, let the loss function $\hat{\mathcal{L}}_u$ be $L-$smooth and let the samples on all agents be drawn from the same distribution $\mathcal{D}$. Let $\mathbf{E}_{\mathcal{D}}[\hat{\mathcal{L}}_u] = \mathcal{L}_u$, and $\Delta_b^t = \hat{\mathcal{L}}_z(\dot{\theta}_b^t) - \hat{\mathcal{L}}_z(\bar{\theta}_b^t)$. Let the algorithm run until round $T$ with step size $\eta \in (0, \frac{1}{L})$, and let $\delta_b := \min_{t \in [T]} \mathbf{E}[\Delta_b^t]$ and $\bar{g}_b^2 := \min_{t \in [T]} \mathbf{E}[\|\nabla \hat{\mathcal{L}}_b(\theta_b^t)\|_2^2]$. Then we have the following results,*
   *a) (Always Trade) If $\Delta_b^t > 0, \forall t$, and agent $B$ always buys (i.e. $I_b^t = 1, \forall t$). Then $T \geq \frac{2(\mathcal{L}(\theta_b^0) - \mathcal{L}(\theta_b^*))}{\eta \epsilon^2 + 2\delta_b}$ implies $\bar{g}_b \leq \epsilon$.*
   *b) (Never Trade) If the agents never trade i.e. ($I_a^t = I_b^t = 0, \forall t$). Then $\bar{g}_b \leq \epsilon$ for $T \geq \frac{2(\mathcal{L}(\theta_b^0) - \mathcal{L}(\theta_b^*))}{\eta \epsilon^2}$.*

**Discussion.** We show the convergence rate of always trade for agent $B$ is $\mathcal{O}(1/(\epsilon^2 + \delta_b))$, while never trade is $\mathcal{O}(1/\epsilon^2)$. The difference between these two scenarios is due to $\delta_b$— the minimal gain-from-trade over $T$ runs. More gain-from-trade implies a smaller $T$ (i.e. faster convergence) in the worst-case scenario. In addition, when $\delta_b$ is $\Omega(\epsilon)$, we can get much better convergence rate of $\mathcal{O}(1/\epsilon)$. Our results in this fundamental setting illustrate that participation in the market can lead to faster convergence when there exists gain-from-trade. The proof for Theorem 5.1 is in the Appendix D.1. In addition to this general setting, we provide convergence analysis for the linear model that we used as a case study in Sec. 4. This analysis also shows a better convergence rate when agents trade parameters. See Appendix D.2 for more details.

# 6 Experiments

We study the proposed framework empirically. Our goals are to validate (i) trading in the proposed framework results in improvements, (ii) these improvements persist even when trading subsets of parameters, (iii) these persist even when agents are trading different models for different tasks, (iv) trading and pricing are viable in competitive settings, and (v) understand the importance of key components of our framework, such as the need for alignment.

## 6.1 Collaborative Agents

We first conduct experiments in a collaborative setting, where there is no payment for buying parameters. Our goal is to validate the potential improvements from transactions independently of the pricing mechanism, which we explore in a later set of experiments.

### 6.1.1 Parameter Trading in Neural Networks

**Setup.** We use MNIST [33], CIFAR10 [34], and TinyImageNet [35] for training MLPs and ResNet20 [36]. Agents have imbalanced datasets where half of the classes contain only 10% of datapoints. Agents are limited to collecting a part of a dataset, making it difficult for them to achieve satisfactory performance without collaborating and trading parameters to leverage each other's strengths.

Models are trained from different random initializations and batch orders over 60 epochs. Agents trade entire parameter sets and join the market after five epochs. The broker discloses gain-from-trade to agents. Broker aligns parameters [16], then merge.

|  |  | Agent $A$ | Agent $B$ |
| --- | --- | --- | --- |
|  |  | Testing Acc. (%) | Testing Acc. (%) |
| MNIST +
MLP | out-of-market | 68.50% | 72.97% |
|  | FedAvg | 81.98% | 81.98% |
|  | w/o alignment | 84.64% | 84.64% |
|  | w alignment | **86.96%** | **86.96%** |
| CIFAR10 +
ResNet20 | out-of-market | 71.14% | 70.56% |
|  | FedAvg | 70.35% | 67.85% |
|  | w/o alignment | 78.31% | 78.31% |
|  | w alignment | **79.90%** | **79.90%** |
| TinyImageNet +
ResNet20 | out-of-market | 21.67% | 15.89% |
|  | FedAvg | 19.95% | 19.33% |
|  | w/o alignment | 31.28% | 31.30% |
|  | w alignment | **31.69%** | **31.82%** |

Table 1: Testing accuracies are reported for each combination of dataset and model. In *FedAvg*, there is no broker to assist agents in conducting transactions. The interpolated weight is determined solely based on the proportion of data assets. *w/o alignment* indicates that broker merges parameters via simple interpolation with the optimized purchased weight. In *w alignment*, the broker aligns parameters by applying [16] and then interpolates.

In addition to the approach that agents train models on their own, we include another baseline method, FedAvg [37], which assumes that there is no broker involved in a trade to help agents align parameters and optimize their purchased weights. In FedAvg, the interpolated weight is determined by the portion of data assets that an agent is endowed with, which is 0.5 in this setting.

**Results.** Table 1 shows the performance of two agents. We find that both agents are able to achieve improved performance by leveraging each other's training expertise, compared to out-of-market agents. Specifically, training and trading ResNet20 with TinyImageNet resulted in improving accuracy by **+10.02%** and **+15.93%**, respectively. We measure two ways to merge parameters for buying: with and without model alignment. With model alignment, the broker is able to merge models more effectively, ultimately enhancing performance for both agents. In addition, compared to FedAvg method, results confirm the significance of having a trusted broker in parameter trading. Without an intermediary broker to facilitate the trade, the performance of purchased weights can be negatively impacted, as evidenced by the results of CIFAR10 + ResNet20 and TinyImageNet + ResNet20.

Finally, Figure 2 displays a comparison of testing loss for the MLP on MNIST. Our results demonstrate that trading parameters results in faster convergence compared to siloed training. Using alignment further helps, as expected.

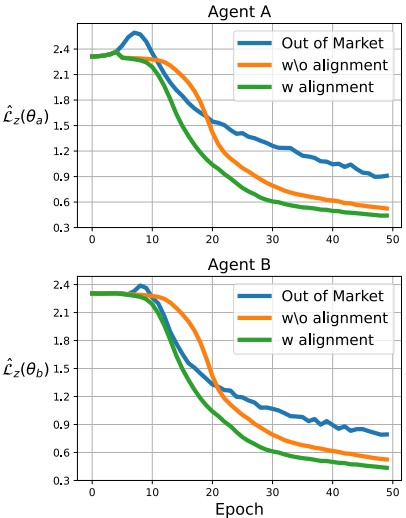

Figure 2: Testing loss converges the fastest by aligning and interpolating.

### 6.1.2 Parameter Subsets Trading in Neural Networks

**Setup.** Next, we explore the potential benefits of trading *subsets* of parameters. This scenario may take place when agents are interested in only certain components or are restricted by trading budgets. We use the same data endowment as in the preceding configuration. We train two 5-layer MLPs on MNIST and align parameters in each layer to trade.

**Results.** Table 2 displays the results by trading parameters from different layers. As expected, trading the entire model gives optimal performance (the last row). Trading subsets is helpful but suboptimal. We observe that purchasing parameters from shallow layers, close to the input layers,

|  | Agent $A$ | Agent $B$ |
|---|---|---|
|  | Testing Acc. (%) | Testing Acc. (%) |
| out-of-market | 71.29% | 72.54% |
| layers $\{3, 4\}$ | 66.28% | 71.98% |
| layers $\{2, 3, 4\}$ | 70.73% | 73.36% |
| layers $\{0, 1, 2\}$ | 74.76% | 74.16% |
| layers $\{0, 1\}$ | 78.86% | 79.82% |
| layers $\{0, 1, 2, 3, 4\}$ | **86.96%** | **86.96%** |

Table 2: Testing accuracies for trading parameters from different layers in 5-layer MLPs on MNIST.

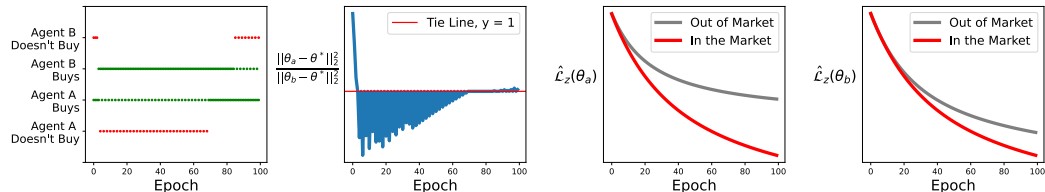

Figure 3: **Both agents earn improvements mutually through trading in the market**. We visualize trading results of two linear regression models with a synthesized dataset. Leftmost: trading logs over 100 runs. Second from left: The ratio of squared parameter estimation error between agent $A$ and agent $B$. A red line represents tied performance. Above the red line, agent $B$ leads, and vice versa. Second from right & rightmost: agent $A$'s and agent $B$'s learning curve compared to out-of-market agents. Market usage thus produces performance improvements.

offers more benefits than trading with deeper layers. This provides trading guidance for budget-limited agents.

### 6.1.3 The Effectiveness of Buying Parameters in Controlled Settings

**Setup.** We use a synthetic dataset to study trading in a fully-controlled environment. We use two linear regression models with dimension $d = 1000$. Agent $A$ has $n_a = 500$ datapoints in dataset $(X_a, Y_a)$ and $B$ has $n_b = 800$ datapoints in dataset $(X_b, Y_b)$, but the latter's labels are affected by zero-mean Gaussian noise ($\sigma^2 = 0.5$). We assume that the broker knows the true parameter $\theta^*$ and obtains $(X_z, Y_z)$ and has access to $n_z = 10,000$ datapoints. Both agents start learning function $f_a, f_b$ with the same initialized weight $\theta^0$. We compare the results over 100 runs with agents who obtain the same data endowment but do not participate in the market.

**Results.** The leftmost Figure 3 displays the trading log over 100 runs. Green dots and red dots show whether an agent purchases the other's parameters in a specific run. Next, we show the ratio of squared parameter estimation error between agents. If the ratio is larger than 1 (red dashed line), agent $B$ leads the market. We see that agent $A$ leads the market in the first half of runs, making agent $B$ continue buying parameters from agent $A$. At the end of a few runs, agent $B$ turns to the lead.

The rightmost plots show convergence. If an agent is involved in the market and trades with the other, the convergence rate is faster compared to never trading. This study demonstrates agents' behaviors in the trading runs and validates the effectiveness of buying parameters, leading to more efficient model training. We compute the empirical testing loss at the end. Compared to out-of-market agents, we find that agent $A$ and agent $B$ are able to improve testing loss by **42.84%** and **23.88%**, respectively.

### 6.1.4 Trading Parameters of Models with Different Purposes

**Setup.** Next, we validate whether trading makes sense even if agents are training *different models for different purposes*. This scenario is more realistic as it simulates situations where organizations in the market are working on different but potentially related tasks. We model this setting by sweeping the distance between the true parameters $\theta_a^*, \theta_b^*$ of two linear regression models to observe how it impacts the benefits of trading.

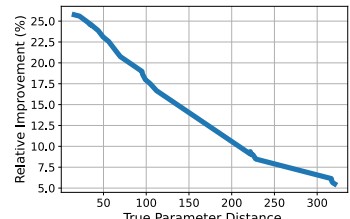

Figure 4: Trading with parameters that are from related tasks is possible.

**Results.** We record the benefits from trading when compared to an agent who does not participate but has the same data as agent

$A$. We measure the relative improvement in empirical testing loss. The results are shown in Figure 4, which indicates that even though the two agents are not training on the same task, agent $A$ is still able to benefit from trading. Note that the gain exists even when the tasks are quite different (i.e. large distance $\|\theta_a^* - \theta_b^*\|_2$) but is strongest as the tasks are most closely related.

## 6.2 Competitive Agents

Finally, we study agents engaging in a competitive scenario. In this case, the transactions require pricing. We validate the proposed pricing mechanism.

### 6.2.1 The Effectiveness of Bayesian Optimal Pricing Mechanism

**Setup.** We reuse the synthesized dataset from the collaborative experiment. We set the buyer's valuation to gain-from-trade and estimate the seller's virtual valuation by the lower bound that we can find.

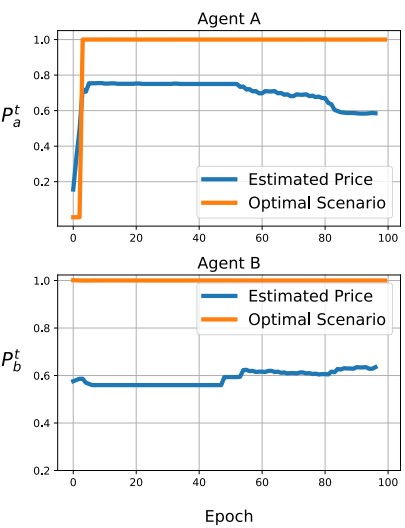

**Results.** Market prices negotiated by the broker over 100 trading runs are displayed in Figure 5. Based on Eq. 1, the market price is determined by the average of the buyer's valuation and the seller's virtual valuation. To demonstrate market efficiency, we also study the scenario where the seller sets the valuation to be exactly the same as the buyer's. As shown in Figure 3 (second from left: performance ratio), agent $A$ is initially leading in the first half of the runs, resulting in a higher price for their parameters. However, at the end of a few runs, agent $B$ takes the lead, causing their parameter price to increase while opponent $A$' goes down. It is important to note that the gap between the estimated price (the blue line) and the optimal price (the orange line) can be reduced with more information learned through the market, such as via historical transactions. Besides, the resulting negotiated price creates a discrepancy with the price in the optimal scenario where both parties report their valuations truthfully, *highlighting the significance of revealing accurate parameter values and justifying the need for incentives.* At last, this study illustrates the feasibility of using the Bayesian optimal pricing mechanism to assist seller agents in monetizing parameters with limited information to make a trade.

Figure 5: We visualize parameter market price negotiated by broker. We can see in the first half of runs, since agent $A$'s performance dominates the market, making agent $A$'s parameter more valuable compared to opponent agent $B$. Note that, if there is no trade, market price remains the same as historical price.

## 7 Conclusion

In this paper, we introduced a framework for parameter markets that can serve to reduce the heavy costs of large-scale model training. Borrowing from economic principles, we provided a set of mechanisms that enable the functioning of such markets. Theoretically, for simple settings, we analyzed market efficiency and proved that agents can gain from participating. We empirically validated the effectiveness of parameter markets under collaborative and competitive settings and demonstrated when participants in the market earn mutual benefits through market usage.

## 8 Acknowledgments

We would like to express our gratitude to the Wisconsin Alumni Research Foundation (WARF) for supporting this work. Additionally, we would like to thank all the reviewers for their valuable comments and constructive feedback on our work.

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
