The appendix is organized as follows. In Appendix A, we show steps for broker to solve revenue maximization problem in a trade (Sec. 3.4). Next, we provide details for the instantiation in Sec. 4. First, in Appendix B, we offer proof for agents to bound performance ratio under different trading decisions. Then, in Appendix C, we provide proof of Theorem 4.1 for agents to bound the other agent's gain-from-trade. We generalize instantiation to a broader setting for L-smooth function and study its convergence analysis in Appendix D.1. In addition to L-smooth function, we use linear models as a case study and provide its convergence analysis in Appendix D.2. In Appendix E, we place experimental details regarding data endowments and training settings. Then, in Appendix F, we offer more findings as trading guidance investigated by different trading scenarios. At last, we discuss limitations, potential concerns, computational needs for broker, and agents' common priors in Appendix G.

## A  Pricing Mechanism

In this section, we offer steps to maximize revenue of both agents. Recall that we define agent $u$'s valuation as $v_u(\theta)$ and the market price of agent $u$'s parameters at time $t$ as $P_u^t$. We set agents' revenue as follows,

$$U_a(P_a^t, P_b^t) := \left(P_a^t - v_a(\dot{\theta}_a^t)\right) + \left(v_a(\dot{\theta}_b^t) - P_b^t\right) \quad U_b(P_b^t, P_a^t) := \left(P_b^t - v_b(\dot{\theta}_b^t)\right) + \left(v_b(\dot{\theta}_a^t) - P_a^t\right)$$

We formulate the revenue maximization problem accordingly,

$$\operatorname*{argmax}_{P_a^t, P_b^t} \quad U_a(P_a^t, P_b^t) \times U_b(P_b^t, P_a^t)$$

$$\text{s.t.} \qquad P_a^t \in \left[v_a(\dot{\theta}_a^t), v_b(\dot{\theta}_a^t)\right], \quad P_b^t \in \left[v_b(\dot{\theta}_b^t), v_b(\dot{\theta}_a^t)\right]$$

Let the difference in price that broker finds as $\Delta P_{ab}^t := P_a^t - P_b^t$. Then we can have,

$$\operatorname*{argmax}_{P_a^t, P_b^t} \left[\left(P_a^t - v_a(\dot{\theta}_a^t)\right) + \left(v_a(\dot{\theta}_b^t) - P_b^t\right)\right] \times \left[\left(P_b^t - v_b(\dot{\theta}_b^t)\right) + \left(v_b(\dot{\theta}_a^t) - P_a^t\right)\right]$$

$$\Rightarrow \operatorname*{argmax}_{P_a^t, P_b^t} \left(\Delta P_{ab}^t - v_a(\dot{\theta}_a^t) + v_a(\dot{\theta}_b^t)\right) \times \left(-\Delta P_{ab}^t - v_b(\dot{\theta}_b^t) + v_b(\dot{\theta}_a^t)\right)$$

$$= \operatorname*{argmax}_{P_a^t, P_b^t} \left(-(\Delta P_{ab}^t)^2 - v_b(\dot{\theta}_b^t) \cdot \Delta P_{ab}^t + v_b(\dot{\theta}_a^t) \cdot \Delta P_{ab}^t + v_a(\dot{\theta}_a^t) \cdot \Delta P_{ab}^t + v_a(\dot{\theta}_a^t) \cdot v_b(\dot{\theta}_b^t)\right.$$

$$\left. - v_a(\dot{\theta}_a^t) \cdot v_b(\dot{\theta}_a^t) - v_a(\dot{\theta}_b^t) \cdot \Delta P_{ab}^t - v_a(\dot{\theta}_b^t) \cdot v_b(\dot{\theta}_b^t) + v_a(\dot{\theta}_b^t) \cdot v_b(\dot{\theta}_a^t)\right)$$

Taking the derivative and set equation to zero,

$$-2\Delta P_{ab}^t - v_b(\dot{\theta}_b^t) + v_b(\dot{\theta}_a^t) + v_a(\dot{\theta}_a^t) - v_a(\dot{\theta}_b^t) = 0$$

Then we can find that the $\Delta P_{ab}^t$ by

$$\Delta P_{ab}^t = P_a^t - P_b^t = \frac{1}{2}\left(v_b(\dot{\theta}_a^t) + v_a(\dot{\theta}_a^t) - v_a(\dot{\theta}_b^t) - v_b(\dot{\theta}_b^t)\right)$$

The resulting price difference represents the transferred payment between agents.

## B  Performance Ratio

We provide a concrete example of how agents value parameters in the linear model in Sec. 4. To approach Theorem 4.1, we start by finding bounds on the performance ratio between agents under different trading decisions. Here, the performance ratio in the market at time $t$ is determined by the following, and we use agent $A$ as an example to bound the ratio between agent $A$ and his opponent.

$$\frac{\|\theta_b^t - \theta^*\|_2^2}{\|\theta_a^t - \theta^*\|_2^2}$$

Once the ratio is greater than 1, agent $A$ is the lead in the market. Note that $\theta_a^t$ and $\theta_b^t$ are the final parameters based on agents' decisions, and $\theta^*$ is the true parameter known by broker. In a two-agent market, There are four different scenarios to analyze.

$$\{\theta_a^t, \theta_b^t\} = \begin{cases} \dot{\theta}_a^t, \dot{\theta}_b^t & \text{if agent } A \text{ doesn't buy and doesn't sell parameters,} \\ \bar{\theta}_a^t, \dot{\theta}_b^t & \text{if agent } A \text{ buys but doesn't sell parameters,} \\ \dot{\theta}_a^t, \bar{\theta}_b^t & \text{if agent } A \text{ doesn't buy but sell parameters,} \\ \bar{\theta}_a^t, \bar{\theta}_b^t & \text{if agent } A \text{ buys and sells parameters.} \end{cases}$$

Recall that the purchased parameters are defined as

$$\bar{\theta}_a^t := (1-\alpha)\dot{\theta}_a^t + \alpha\dot{\theta}_b^t, \quad \alpha \in (0,1]$$
$$\bar{\theta}_b^t := (1-\beta)\dot{\theta}_b^t + \beta\dot{\theta}_a^t, \quad \beta \in (0,1]$$

In the instantiation, recall that we take the gain-from-trade revealed by broker for agent $A$ to be

$$\Delta_a^t := \frac{\|\dot{\theta}_a^t - \theta^*\|_2^2}{\|\bar{\theta}_a^t - \theta^*\|_2^2}$$

**Lemma B.1.** *Agent A: (No-Buy, No-Sell) If agent $A$ doesn't buy parameters from agent $B$ and doesn't sell parameters to agent $B$, then knowing the purchased weight $\alpha$ and gain-from-trade $\Delta_a^t$, the ratio of parameter estimation error between agent $A$ and agent $B$ is bounded by:*

$$\left(\frac{1}{\alpha\sqrt{\Delta_a^t}} - \frac{1-\alpha}{\alpha}\right)^2 \leq \frac{\|\dot{\theta}_b^t - \theta^*\|_2^2}{\|\dot{\theta}_a^t - \theta^*\|_2^2} \leq \left(\frac{1}{\alpha\sqrt{\Delta_a^t}} + \frac{1-\alpha}{\alpha}\right)^2$$

**Lemma B.2.** *Agent A: (Buy, No-Sell) If agent $A$ buys parameters from agent $B$ and doesn't sell parameters to agent $B$, then knowing the purchased weight $\alpha$ and gain-from-trade $\Delta_a^t$, the ratio of parameter estimation error between agent $A$ and agent $B$ is bounded by:*

$$\Delta_a^t\left(\frac{1}{\alpha\sqrt{\Delta_a^t}} - \frac{1-\alpha}{\alpha}\right)^2 \leq \frac{\|\dot{\theta}_b^t - \theta^*\|_2^2}{\|\bar{\theta}_a^t - \theta^*\|_2^2} \leq \Delta_a^t\left(\frac{1}{\alpha\sqrt{\Delta_a^t}} + \frac{1-\alpha}{\alpha}\right)^2$$

*Proof.*

$$\begin{aligned}
\|\dot{\theta}_b^t - \theta^*\|_2^2 &= \|\frac{\bar{\theta}_a^t - (1-\alpha)\dot{\theta}_a^t}{\alpha} - \theta^*\|_2^2 \\
&= \|\frac{1}{\alpha}(\bar{\theta}_a^t - \theta^*) - \frac{1-\alpha}{\alpha}(\dot{\theta}_a^t - \theta^*)\|_2^2 \\
&= \frac{1}{\alpha^2}\|\bar{\theta}_a^t - \theta^*\|_2^2 + (\frac{1-\alpha}{\alpha})^2\|\dot{\theta}_a^t - \theta^*\|_2^2 - \frac{2(1-\alpha)}{\alpha^2}\langle\bar{\theta}_a^t - \theta^*, \dot{\theta}_a^t - \theta^*\rangle \\
&\leq \frac{1}{\alpha^2}\|\bar{\theta}_a^t - \theta^*\|_2^2 + (\frac{1-\alpha}{\alpha})^2\|\dot{\theta}_a^t - \theta^*\|_2^2 + \frac{2(1-\alpha)}{\alpha^2}\|\bar{\theta}_a^t - \theta^*\|_2\|\dot{\theta}_a^t - \theta^*\|_2 \\
&= \frac{1}{\Delta_a^t\alpha^2}\|\dot{\theta}_a^t - \theta^*\|_2^2 + (\frac{1-\alpha}{\alpha})^2\|\dot{\theta}_a^t - \theta^*\|_2^2 + \frac{2(1-\alpha)}{\sqrt{\Delta_a^t}\alpha^2}\|\dot{\theta}_a^t - \theta^*\|_2^2 \\
&= \left(\frac{1}{\alpha\sqrt{\Delta_a^t}} + \frac{1-\alpha}{\alpha}\right)^2\|\dot{\theta}_a^t - \theta^*\|_2^2
\end{aligned}$$

Hence, we can have an upper bound

$$\frac{\|\dot{\theta}_b^t - \theta^*\|_2^2}{\|\dot{\theta}_a^t - \theta^*\|_2^2} \leq \left(\frac{1}{\alpha\sqrt{\Delta_a^t}} + \frac{1-\alpha}{\alpha}\right)^2$$

For the lower bound,

$$
\begin{aligned}
\|\dot{\theta}_b^t - \theta^*\|_2^2 &= \|\frac{\bar{\theta}_a^t - (1-\alpha)\dot{\theta}_a^t}{\alpha} - \theta^*\|_2^2 \\
&= \|\frac{1}{\alpha}(\bar{\theta}_a^t - \theta^*) - \frac{1-\alpha}{\alpha}(\dot{\theta}_a^t - \theta^*)\|_2^2 \\
&= \frac{1}{\alpha^2}\|\bar{\theta}_a^t - \theta^*\|_2^2 + (\frac{1-\alpha}{\alpha})^2\|\dot{\theta}_a^t - \theta^*\|_2^2 - \frac{2(1-\alpha)}{\alpha^2}\langle\bar{\theta}_a^t - \theta^*, \dot{\theta}_a^t - \theta^*\rangle \\
&\geq \frac{1}{\alpha^2}\|\bar{\theta}_a^t - \theta^*\|_2^2 + (\frac{1-\alpha}{\alpha})^2\|\dot{\theta}_a^t - \theta^*\|_2^2 - \frac{2(1-\alpha)}{\alpha^2}\|\bar{\theta}_a^t - \theta^*\|_2\|\dot{\theta}_a^t - \theta^*\|_2 \\
&= \frac{1}{\Delta_a^t\alpha^2}\|\dot{\theta}_a^t - \theta^*\|_2^2 + (\frac{1-\alpha}{\alpha})^2\|\dot{\theta}_a^t - \theta^*\|_2^2 - \frac{2(1-\alpha)}{\sqrt{\Delta_a^t}\alpha^2}\|\dot{\theta}_a^t - \theta^*\|_2^2 \\
&= (\frac{1}{\alpha\sqrt{\Delta_a^t}} - \frac{1-\alpha}{\alpha})^2\|\dot{\theta}_a^t - \theta^*\|_2^2
\end{aligned}
$$

Hence, we can have a lower bound satisfied by

$$
\frac{\|\dot{\theta}_b^t - \theta^*\|_2^2}{\|\dot{\theta}_a^t - \theta^*\|_2^2} \geq (\frac{1}{\alpha\sqrt{\Delta_a^t}} - \frac{1-\alpha}{\alpha})^2
$$

Therefore, by knowing gain-from-trade $\Delta_a^t$ and purchased weight $\alpha$ from the broker, in {No-Buy, No-Sell} scenario, agent $A$ can bound the performance ratio with the final parameter $\dot{\theta}_a^t$ and $\dot{\theta}_b^t$. We can also have bounds for another scenario {Buy, No-Sell} by multiplying $\Delta_a^t$ to the inequality. $\quad\square$

Another case in the market is agent $B$ wishes to buy parameters from agent $A$. In this case, agent $A$ will receive a quotation request from agent $B$ with his purchased weight $\beta$. Then, agent $A$ is able to bound the performance ratio.

**Lemma B.3.** *Agent A: (No-Buy, Sell) If agent A doesn't buy parameters from agent B but sells parameters to agent B, then knowing purchased weights $\alpha, \beta$ and gain-from-trade $\Delta_a^t$, the ratio of parameter estimation error between agent A and agent B is bounded by:*

$$
\left[(1-\beta)(\frac{1}{\alpha\sqrt{\Delta_a^t}} - \frac{1-\alpha}{\alpha}) - \beta\right]^2 \leq \frac{\|\bar{\theta}_b^t - \theta^*\|_2^2}{\|\dot{\theta}_a^t - \theta^*\|_2^2} \leq \left[(1-\beta)(\frac{1}{\alpha\sqrt{\Delta_a^t}} + \frac{1-\alpha}{\alpha}) + \beta\right]^2
$$

**Lemma B.4.** *Agent A: (Buy, Sell) If agent A buys parameters from agent B and sells parameters to agent B, then knowing purchased weights $\alpha, \beta$ and gain-from-trade $\Delta_a^t$, the ratio of parameter estimation error between agent A and agent B is bounded by:*

$$
\Delta_a^t\left[(1-\beta)(\frac{1}{\alpha\sqrt{\Delta_a^t}} - \frac{1-\alpha}{\alpha}) - \beta\right]^2 \leq \frac{\|\bar{\theta}_b^t - \theta^*\|_2^2}{\|\bar{\theta}_a^t - \theta^*\|_2^2} \leq \Delta_a^t\left[(1-\beta)(\frac{1}{\alpha\sqrt{\Delta_a^t}} + \frac{1-\alpha}{\alpha}) + \beta\right]^2
$$

*Proof.*

$$
\begin{aligned}
\|\bar{\theta}_b^t - \theta^*\|_2^2 &= \|(1-\beta)\dot{\theta}_b^t + \beta\dot{\theta}_a^t - \theta^*\|_2^2 \\
&= \|(1-\beta)(\dot{\theta}_b^t - \theta^*) + \beta(\dot{\theta}_a^t - \theta^*)\|_2^2 \\
&= (1-\beta)^2\|\dot{\theta}_b^t - \theta^*\|_2^2 + \beta^2\|\dot{\theta}_a^t - \theta^*\|_2^2 + 2(1-\beta)\beta\langle\dot{\theta}_b^t - \theta^*, \dot{\theta}_a^t - \theta^*\rangle \\
&\leq (1-\beta)^2\|\dot{\theta}_b^t - \theta^*\|_2^2 + \beta^2\|\dot{\theta}_a^t - \theta^*\|_2^2 + 2(1-\beta)\beta\|\dot{\theta}_b^t - \theta^*\|_2\|\dot{\theta}_a^t - \theta^*\|_2 \\
&\leq (1-\beta)^2(\frac{1}{\alpha\sqrt{\Delta_a^t}} + \frac{1-\alpha}{\alpha})^2\|\dot{\theta}_a^t - \theta^*\|_2^2 + \beta^2\|\dot{\theta}_a^t - \theta^*\|_2^2 \\
&\quad + 2(1-\beta)\beta(\frac{1}{\alpha\sqrt{\Delta_a^t}} + \frac{1-\alpha}{\alpha})\|\dot{\theta}_a^t - \theta^*\|_2^2 \qquad\qquad\text{(Lemma B.1)} \\
&= \left[(1-\beta)(\frac{1}{\alpha\sqrt{\Delta_a^t}} + \frac{1-\alpha}{\alpha}) + \beta\right]^2\|\dot{\theta}_a^t - \theta^*\|_2^2
\end{aligned}
$$

Hence, we can have an upper bound

$$\frac{\|\bar{\theta}_b^t - \theta^*\|_2^2}{\|\dot{\theta}_a^t - \theta^*\|_2^2} \leq \left[(1-\beta)\left(\frac{1}{\alpha\sqrt{\Delta_a^t}} + \frac{1-\alpha}{\alpha}\right) + \beta\right]^2$$

For the lower bound,

$$\begin{aligned}
\|\bar{\theta}_b^t - \theta^*\|_2^2 &= \|(1-\beta)\dot{\theta}_b^t + \beta\dot{\theta}_a^t - \theta^*\|_2^2 \\
&= \|(1-\beta)(\dot{\theta}_b^t - \theta^*) + \beta(\dot{\theta}_a^t - \theta^*)\|_2^2 \\
&= (1-\beta)^2\|\dot{\theta}_b^t - \theta^*\|_2^2 + \beta^2\|\dot{\theta}_a^t - \theta^*\|_2^2 + 2(1-\beta)\beta\langle\dot{\theta}_b^t - \theta^*, \dot{\theta}_a^t - \theta^*\rangle \\
&\geq (1-\beta)^2\|\dot{\theta}_b^t - \theta^*\|_2^2 + \beta^2\|\dot{\theta}_a^t - \theta^*\|_2^2 - 2(1-\beta)\beta\|\dot{\theta}_b^t - \theta^*\|_2\|\dot{\theta}_a^t - \theta^*\|_2 \\
&\geq (1-\beta)^2\left(\frac{1}{\alpha\sqrt{\Delta_a^t}} - \frac{1-\alpha}{\alpha}\right)^2\|\dot{\theta}_a^t - \theta^*\|_2^2 + \beta^2\|\dot{\theta}_a^t - \theta^*\|_2^2 \\
&\quad - 2(1-\beta)\beta\left(\frac{1}{\alpha\sqrt{\Delta_a^t}} - \frac{1-\alpha}{\alpha}\right)\|\dot{\theta}_a^t - \theta^*\|_2^2 \qquad\qquad \text{(Lemma B.1)} \\
&= \left[(1-\beta)\left(\frac{1}{\alpha\sqrt{\Delta_a^t}} - \frac{1-\alpha}{\alpha}\right) - \beta\right]^2\|\dot{\theta}_a^t - \theta^*\|_2^2
\end{aligned}$$

Hence, we can have a lower bound

$$\frac{\|\bar{\theta}_b^t - \theta^*\|_2^2}{\|\dot{\theta}_a^t - \theta^*\|_2^2} \geq \left[(1-\beta)\left(\frac{1}{\alpha\sqrt{\Delta_a^t}} - \frac{1-\alpha}{\alpha}\right) - \beta\right]^2$$

Therefore, by knowing gain-from-trade $\Delta_a^t$ and purchased weight $\alpha, \beta$, in {No-Buy, Sell} scenario, agent $A$ can bound the performance ratio with the final parameter $\dot{\theta}_a^t$ and $\bar{\theta}_b^t$. We can also have bounds for another scenario {Buy, Sell} by multiplying $\Delta_a^t$ to the inequality. $\qquad\square$

## C  Buyer's Gain-from-Trade

Next, we use Lemma B.1 and Lemma B.3 to approach Theorem 4.1. Recall that agent $B$'s gain-from-trade $\Delta_b^t$ at the time $t$ is defined as,

$$\Delta_b^t := \frac{\|\dot{\theta}_b^t - \theta^*\|_2^2}{\|\bar{\theta}_b^t - \theta^*\|_2^2}$$

**Theorem C.1.** ***Bounds on Buyer's Gain-from-Trade****: By knowing purchased weights $\alpha$, $\beta$ and $\Delta_a^t$, agent $A$ can obtain bounds on the gain-from-trade of agent $B$ given by:*

$$\left(\frac{1 - \sqrt{\Delta_a^t}(1-\alpha)}{(1-\beta) + \sqrt{\Delta_a^t}(1-\alpha-\beta+2\alpha\beta)}\right)^2 \leq \Delta_b^t \leq \left(\frac{1 + \sqrt{\Delta_a^t}(1-\alpha)}{(1-\beta) - \sqrt{\Delta_a^t}(1-\alpha-\beta+2\alpha\beta)}\right)^2.$$

*Proof.* Based on Lemma B.1, we have

$$\left(\frac{1}{\alpha\sqrt{\Delta_a^t}} - \frac{1-\alpha}{\alpha}\right)^2\|\dot{\theta}_a^t - \theta^*\|_2^2 \leq \|\dot{\theta}_b^t - \theta^*\|_2^2 \leq \left(\frac{1}{\alpha\sqrt{\Delta_a^t}} + \frac{1-\alpha}{\alpha}\right)^2\|\dot{\theta}_a^t - \theta^*\|_2^2$$

Dividing by $\|\bar{\theta}_b^t - \theta^*\|_2^2$, we can have

$$\left(\frac{1}{\alpha\sqrt{\Delta_a^t}} - \frac{1-\alpha}{\alpha}\right)^2\frac{\|\dot{\theta}_a^t - \theta^*\|_2^2}{\|\bar{\theta}_b^t - \theta^*\|_2^2} \leq \Delta_b^t \leq \left(\frac{1}{\alpha\sqrt{\Delta_a^t}} + \frac{1-\alpha}{\alpha}\right)^2\frac{\|\dot{\theta}_a^t - \theta^*\|_2^2}{\|\bar{\theta}_b^t - \theta^*\|_2^2}$$

Since we know Lemma B.3 which gives us

$$\left[(1-\beta)\left(\frac{1}{\alpha\sqrt{\Delta_a^t}} - \frac{1-\alpha}{\alpha}\right) - \beta\right]^2 \leq \frac{\|\bar{\theta}_b^t - \theta^*\|_2^2}{\|\dot{\theta}_a^t - \theta^*\|_2^2} \leq \left[(1-\beta)\left(\frac{1}{\alpha\sqrt{\Delta_a^t}} + \frac{1-\alpha}{\alpha}\right) + \beta\right]^2$$

Inverting the fraction, we can have

$$\left(\frac{\alpha\sqrt{\Delta_a^t}}{(1-\beta)+\sqrt{\Delta_a^t(1-\alpha-\beta+2\alpha\beta)}}\right)^2 \le \frac{\|\dot{\theta}_a^t-\theta^*\|_2^2}{\|\bar{\theta}_b^t-\theta^*\|_2^2} \le \left(\frac{\alpha\sqrt{\Delta_a^t}}{(1-\beta)-\sqrt{\Delta_a^t(1-\alpha-\beta+2\alpha\beta)}}\right)^2$$

Therefore, using Lemma B.1, the upper bound of $\Delta_b^t$ can be found by

$$\Delta_b^t \le (\frac{1}{\alpha\sqrt{\Delta_a^t}}+\frac{1-\alpha}{\alpha})^2(\frac{\alpha\sqrt{\Delta_a^t}}{(1-\beta)-\sqrt{\Delta_a^t(1-\alpha-\beta+2\alpha\beta)}})^2$$

$$\le \left(\frac{1+\sqrt{\Delta_a^t}(1-\alpha)}{(1-\beta)-\sqrt{\Delta_a^t(1-\alpha-\beta+2\alpha\beta)}}\right)^2$$

On the other side, we can have the lower bound as follows

$$\Delta_b^t \ge (\frac{1}{\alpha\sqrt{\Delta_a^t}}-\frac{1-\alpha}{\alpha})^2(\frac{\alpha\sqrt{\Delta_a^t}}{(1-\beta)+\sqrt{\Delta_a^t(1-\alpha-\beta+2\alpha\beta)}})^2$$

$$\ge \left(\frac{1-\sqrt{\Delta_a^t}(1-\alpha)}{(1-\beta)+\sqrt{\Delta_a^t(1-\alpha-\beta+2\alpha\beta)}}\right)^2$$

Theorem 4.1 asserts that if the broker discloses certain information; purchased weights $\alpha, \beta$ and gain-from-trade $\Delta_a^t$, agent $A$ as a seller can determine the maximum and minimum values of buyer's gain-from-trade, $\Delta_b^t$. This knowledge can then be utilized to approximate buyer's valuation so that seller's virtual valuation can be set. $\qquad\square$

# D   Convergence Analysis with Trading

## D.1   (General) L-smooth Functions

In Sec. 5, we study a broader setting for general L-smooth functions and offer its convergence analysis to validate the *effectiveness of buying parameters*. In the general case, the broker informs the gain-from-trade by using the subtraction of empirical loss before and after a trade. This can be more practical, as the broker doesn't need to be knowledgeable about the true parameter $\theta^*$. We write the gain-from-trade as follows,

$$\Delta_u^t := \hat{\mathcal{L}}_z(\dot{\theta}_u^t) - \hat{\mathcal{L}}_z(\bar{\theta}_u^t)$$

**Theorem D.1.** *For all agents $u \in \{a, b, z\}$, let the loss function $\hat{\mathcal{L}}_u$ be $L-$smooth, and let the samples on all agents be drawn from the same distribution $\mathcal{D}$. Let $\mathbf{E}_{\mathcal{D}}[\hat{\mathcal{L}}_u] = \mathcal{L}_u$, and $\Delta_b^t = \hat{\mathcal{L}}_z(\dot{\theta}_b^t) - \hat{\mathcal{L}}_z(\bar{\theta}_b^t)$. Let the algorithm run until round $T$ with step size $\eta \in (0, \frac{1}{L})$, and let $\delta_b := \min_{t \in [T]} \mathbf{E}[\Delta_b^t]$ and $\bar{g}_b^2 := \min_{t \in [T]} \mathbf{E}[\|\nabla\hat{\mathcal{L}}_b(\theta_b^t)\|_2^2]$. Then we have the following results,*
  *a) (Always Trade) If $\Delta_b^t > 0, \forall t$, and agent $B$ always buys (i.e. $I_b^t = 1, \forall t$). Then $T \ge \frac{2\left(\mathcal{L}(\theta_b^0)-\mathcal{L}(\theta_b^*)\right)}{\eta\epsilon^2+2\delta_b}$ implies $\bar{g}_b \le \epsilon$.*
  *b) (Never Trade) If the agents never trade i.e. $(I_a^t = I_b^t = 0, \forall t)$. Then $\bar{g}_b \le \epsilon$ for $T \ge \frac{2\left(\mathcal{L}(\theta_b^0)-\mathcal{L}(\theta_b^*)\right)}{\eta\epsilon^2}$.*

*Proof.* The proof for the never trade case follows from standard analysis of gradient descent for L-smooth functions (see the proof of Proposition D.2). Here we provide proof for the always trade case. In this case we have $\Delta_b^t > 0$ for all rounds, causing agent $B$ always buy parameters. The population level loss function is defined as $\mathcal{L}$, it is the expectation of $\hat{\mathcal{L}}_u, u \in \{a, b, z\}$. Hence, the expectation of the gain-from-trade for agent $B$ can be written as

$$\mathbf{E}[\Delta_b^t] = \mathbf{E}[\hat{\mathcal{L}}_b(\dot{\theta}_b^t)] - \mathbf{E}[\hat{\mathcal{L}}_b(\bar{\theta}_b^t)] = \mathcal{L}(\dot{\theta}_b^t) - \mathcal{L}(\bar{\theta}_b^t) \tag{5}$$

Since the loss function, $\hat{\mathcal{L}}$ is L-smooth, thus we have the following descent lemma (see Eq. Descent Lemma in the proof of Proposition D.2),

$$\mathcal{L}(\dot{\theta}_b^t) \leq \mathcal{L}(\theta_b^{t-1}) - \frac{\eta}{2}\mathbf{E}[\|\nabla\hat{\mathcal{L}}_b(\theta_b^{t-1})\|_2^2], \quad \forall t$$

Using Eq. 5, we can have $\mathcal{L}(\dot{\theta}_b^t) = \mathcal{L}(\bar{\theta}_b^t) + \mathbf{E}[\Delta_b^t]$, substituting it in the above descent lemma we get,

$$\mathcal{L}(\bar{\theta}_b^t) \leq \mathcal{L}(\theta_b^{t-1}) - \frac{\eta}{2}\mathbf{E}[\|\nabla\hat{\mathcal{L}}_b(\theta_b^{t-1})\|_2^2] - \mathbf{E}[\Delta_b^t], \quad \forall t \tag{6}$$

Note that, since agent $B$ always purchases parameters, the final parameter $\theta_b^t$ is exactly as same as $\bar{\theta}_b^t$. Then we sum Eq. 6 over $T$ rounds, we obtain

$$\mathcal{L}(\theta_b^T) = \mathcal{L}(\bar{\theta}_b^T) \leq \mathcal{L}(\theta_b^0) - \frac{\eta}{2}\sum_{i=1}^{T}\mathbf{E}[\|\nabla\hat{\mathcal{L}}_b(\theta_b^{i-1})\|_2^2] - \sum_{i=1}^{T}\mathbf{E}[\Delta_b^i]$$

Let $\delta_b := \min_t \mathbf{E}[\Delta_b^t], \forall t$ and $\bar{g}_b^2 := \min_t \mathbf{E}[\|\nabla\hat{\mathcal{L}}_b(\theta_b^t)\|_2^2]$. Then,

$$\mathcal{L}(\theta_b^*) \leq \mathcal{L}(\theta_b^T) \leq \mathcal{L}(\theta_b^0) - \frac{\eta T}{2}\bar{g}_b^2 - T\delta_b$$

$$\bar{g}_b^2 \leq \frac{2\big(\mathcal{L}(\theta_b^0) - \mathcal{L}(\theta_b^*)\big)}{\eta T} - \frac{2\delta_b}{\eta}$$

Want the R.H.S. to be at most $\epsilon^2$,

$$\frac{2\big(\mathcal{L}(\theta_b^0) - \mathcal{L}(\theta_b^*)\big)}{\eta T} - \frac{2\delta_b}{\eta} \leq \epsilon^2$$
$$2\big(\mathcal{L}(\theta_b^0) - \mathcal{L}(\theta_b^*)\big) - 2\delta_b T \leq \eta\epsilon^2 T$$
$$T \geq \frac{2(\mathcal{L}(\theta_b^0) - \mathcal{L}(\theta_b^*))}{\eta\epsilon^2 + 2\delta_b}$$

Leading to convergence rate of $\mathcal{O}(1/(\epsilon^2 + \delta_b))$. Additionally, when $\delta_b$ is $\Omega(\epsilon)$, then we get a much better convergence rate of $\mathcal{O}(1/\epsilon)$. □

**Proposition D.2.** *(Never trade, L-smooth Loss) Let the loss functions $\hat{\mathcal{L}}_u$ be L-smooth. Let the samples on all agents be drawn from the same distribution $\mathcal{D}$, and $\mathbf{E}_{\mathcal{D}}[\hat{\mathcal{L}}_u] = \mathcal{L}_u$. Let agent $B$ never trade (i.e. $I_b^t = 0, \forall t$). Let the algorithm run until round $T$ with step size $\eta \in (0, \frac{1}{L})$, and let $\bar{g}_b^2 := \min_{t \in [T]} \mathbf{E}[\|\nabla\hat{\mathcal{L}}_b(\theta_b^t)\|_2^2]$. Then $\bar{g}_b \leq \epsilon$, for*

$$T \geq \frac{2(\mathcal{L}(\theta_b^0) - \mathcal{L}(\theta_b^*))}{\eta\epsilon^2}$$

*Proof.* The proof is a standard analysis of gradient descent for L-smooth functions. We provide the proof here for reference. Due to the L-smoothness of the loss functions we have,

$$\hat{\mathcal{L}}_z(\dot{\theta}_b^t) \leq \hat{\mathcal{L}}_z(\theta_b^{t-1}) + \langle\nabla\hat{\mathcal{L}}_z(\theta_b^{t-1}), \dot{\theta}_b^t - \theta_b^{t-1}\rangle + \frac{L}{2}\|\dot{\theta}_b^t - \theta_b^{t-1}\|_2^2$$

Since, $\dot{\theta}_b^t = \theta_b^{t-1} - \eta\nabla\hat{\mathcal{L}}_z(\theta_b^{t-1})$,

$$\hat{\mathcal{L}}_z(\dot{\theta}_b^t) \leq \hat{\mathcal{L}}_z(\theta_b^{t-1}) - \eta\|\nabla\hat{\mathcal{L}}_z(\theta_b^{t-1})\|_2^2 + \frac{\eta^2 L}{2}\|\nabla\hat{\mathcal{L}}_z(\theta_b^{t-1})\|_2^2$$

$$= \hat{\mathcal{L}}_z(\theta_b^{t-1}) - \Big(\eta - \frac{\eta^2 L}{2}\Big)\|\nabla\hat{\mathcal{L}}_z(\theta_b^{t-1})\|_2^2$$

The constant $\eta - \frac{\eta^2 L}{2}$ is lower bounded by $\eta/2$ for $\eta \in (0, 1/L)$. Leading to the following descent lemma,

$$\hat{\mathcal{L}}_z(\dot{\theta}_b^t) \leq \hat{\mathcal{L}}_z(\theta_b^{t-1}) - \frac{\eta}{2}\|\nabla\hat{\mathcal{L}}_z(\theta_b^{t-1})\|_2^2$$

Taking expectation over $\mathcal{D}$,

$$\mathcal{L}(\dot{\theta}_b^t) \leq \mathcal{L}(\theta_b^{t-1}) - \frac{\eta}{2}\mathbf{E}[\|\nabla\hat{\mathcal{L}}_z(\theta_b^{t-1})\|_2^2] \qquad \text{(Descent Lemma)}$$

Since agent $B$ never trades, making $\theta_b^t = \dot{\theta}_b^t$,

$$\mathcal{L}(\theta_b^t) \leq \mathcal{L}(\theta_b^{t-1}) - \frac{\eta}{2}\mathbf{E}[\|\nabla\hat{\mathcal{L}}_z(\theta_b^{t-1})\|_2^2]$$

Now, summing the above equation of $T$ rounds, we can have

$$\mathcal{L}(\theta^*) \leq \mathcal{L}(\theta_b^T) \leq \mathcal{L}(\theta_b^0) - \frac{\eta}{2}\sum_{t=1}^{T}\mathbf{E}[\|\nabla\hat{\mathcal{L}}_z(\theta_b^{t-1})\|_2^2]$$

$$\sum_{t=1}^{T}\mathbf{E}[\|\nabla\hat{\mathcal{L}}_z(\theta_b^{t-1})\|_2^2] \leq \frac{2(\mathcal{L}(\theta^0) - \mathcal{L}(\theta^*))}{\eta}$$

Since $\bar{g}_b^2 := \min_{t\in[T]}\mathbf{E}[\|\nabla\hat{\mathcal{L}}_b(\theta_b^t)\|_2^2]$,

$$\bar{g}_b^2 \leq \frac{2(\mathcal{L}(\theta^0) - \mathcal{L}(\theta^*))}{\eta T}$$

Solving for $\bar{g}_b \leq \epsilon$ gives us,

$$T \geq \frac{2(\mathcal{L}(\theta^0) - \mathcal{L}(\theta^*))}{\eta\epsilon^2}$$

The convergence rate of *never trade* is $\mathcal{O}(1/\epsilon^2)$, which can be slower than *always trade* $\mathcal{O}\big(1/(\epsilon^2 + \delta_b)\big)$ by the constant factor, gain-from-trade. $\qquad\square$

## D.2 Linear Model Case Study

In addition to general case, we study the convergence for the instantiation in Sec. 4, where agents are trading parameters in linear models, and the gain-from-trade computed by the broker is defined as

$$\Delta_u^t := \frac{\|\dot{\theta}_u^t - \theta^*\|_2^2}{\|\bar{\theta}_u^t - \theta^*\|_2^2}, \quad u \in \{a, b\}$$

In this setting, we write the empirical loss function as

$$\hat{\mathcal{L}}_u(\theta) = \|X_u\theta - Y_u\|_2^2$$

Recall that the updated rules are

$$\dot{\theta}_a^t = \theta_a^{t-1} - \eta\nabla\hat{\mathcal{L}}_a(\theta_a^{t-1}) \qquad\qquad \dot{\theta}_b^t = \theta_b^{t-1} - \eta\nabla\hat{\mathcal{L}}_b(\theta_b^{t-1})$$
$$\bar{\theta}_a^t = (1-\alpha)\cdot\dot{\theta}_a^t + \alpha\cdot\dot{\theta}_b^t \qquad\qquad \bar{\theta}_b^t = (1-\beta)\cdot\dot{\theta}_b^t + \beta\cdot\dot{\theta}_a^t$$
$$\theta_a^t = (1-I_a^t)\cdot\dot{\theta}_a^t + I_a^t\cdot\bar{\theta}_a^t \qquad\qquad \theta_b^t = (1-I_b^t)\cdot\dot{\theta}_b^t + I_b^t\cdot\bar{\theta}_b^t$$

**Lemma D.3.** *Let $X_u^T X_u$ be a p.d. matrix. Let $\lambda_{\max}(X_u^T X_u), \lambda_{\min}(X_u^T X_u)$ be the maximum and minimum eigenvalues of $X_u^T X_u$. Denote the condition number $\rho_u := \frac{\lambda_{\max}(X_u^T X_u)}{\lambda_{\min}(X_u^T X_u)}$, and $Y_u = X_u\theta^*$ then,*

$$\frac{1}{\rho_u\Delta_u^t}\hat{\mathcal{L}}_u(\dot{\theta}_u^t) \leq \hat{\mathcal{L}}_u(\bar{\theta}_u^t) \leq \frac{\rho_u}{\Delta_u^t}\hat{\mathcal{L}}_u(\dot{\theta}_u^t) \qquad\qquad (7)$$

*Proof.* In the noiseless case, $Y_u = X_u \theta^*$, which gives

$$\hat{\mathcal{L}}_u(\theta) = \|X_u \theta - Y_u\|_2^2 = \|X_u \theta - X_u \theta^*\|_2^2 = \|X_u(\theta - \theta^*)\|_2^2$$

Another way to write this is $\hat{\mathcal{L}}_u(\theta) = (\theta - \theta^*)^T X_u^T X_u (\theta - \theta^*)$. Since $X_u^T X_u$ is a positive definite matrix, we have

$$\lambda_{\min}(X_u^T X_u)\|\theta - \theta^*\|_2^2 \leq \hat{\mathcal{L}}_u(\theta) \leq \lambda_{\max}(X_u^T X_u)\|\theta - \theta^*\|_2^2$$

This implies the following

$$\frac{\hat{\mathcal{L}}_u(\dot{\theta}_u^t)}{\hat{\mathcal{L}}_u(\bar{\theta}_u^t)} \leq \frac{\lambda_{\max}(X_u^T X_u)\|\dot{\theta}_u^t - \theta^*\|_2^2}{\lambda_{\min}(X_u^T X_u)\|\bar{\theta}_u^t - \theta^*\|_2^2} = \rho_u \Delta_u^t$$

and

$$\frac{\hat{\mathcal{L}}_u(\dot{\theta}_u^t)}{\hat{\mathcal{L}}_u(\bar{\theta}_u^t)} \geq \frac{\lambda_{\min}(X_u^T X_u)\|\dot{\theta}_u^t - \theta^*\|_2^2}{\lambda_{\max}(X_u^T X_u)\|\bar{\theta}_u^t - \theta^*\|_2^2} = \frac{\Delta_u^t}{\rho_u}$$

$\square$

**Theorem D.4.** *(Always Trade, Linear Case) Let $\Delta_b^t$ be the gain-from-trade of agent $B$ at round $t$. Assume $\Delta_b^t > 1$ for all rounds, making agent $B$ always buy parameters. Let $\delta_b' = \min_t \Delta_b^t$ and $\delta_b'/\rho_b > 1$. Then for any $\epsilon \in (0,1)$ at the end of round $T$, we have $\hat{\mathcal{L}}_z(\theta_b^T) - \hat{\mathcal{L}}_z(\theta^*) \leq \epsilon$, for*

$$T \geq \frac{1}{\log(\delta_b'/\rho_b)} \log\left(\frac{\hat{\mathcal{L}}_b(\theta_b^0) - \hat{\mathcal{L}}_z(\theta^*)}{\epsilon}\right)$$

*Proof.* The proof follows by using Lemma D.3 to establish a recurrence relation on $\hat{\mathcal{L}}_b(\theta_b^t)$ and then using the fact that $\hat{\mathcal{L}}_z(\theta) \leq \hat{\mathcal{L}}_b(\theta), \forall \theta$ gives us the result. The steps are as follows,

$$\hat{\mathcal{L}}_b(\theta_b^t) = \hat{\mathcal{L}}_b(\bar{\theta}_b^t) \leq \frac{\rho_b}{\Delta_b^t} \hat{\mathcal{L}}_b(\dot{\theta}_b^t)$$

$$\leq \frac{\rho_b}{\Delta_b^t}\left(\hat{\mathcal{L}}_b(\theta_b^{t-1}) - \frac{\eta}{2}\|\nabla\hat{\mathcal{L}}_b(\theta_b^{t-1})\|_2^2\right)$$

$$\leq \frac{\rho_b}{\Delta_b^t}\hat{\mathcal{L}}_b(\theta_b^{t-1})$$

The second step follows from the fact that $\dot{\theta}_b^t = \theta_b^{t-1} - \eta\nabla\hat{\mathcal{L}}_b(\theta_b^{t-1})$ and since $\hat{\mathcal{L}}_b$ is L-smooth, this implies the iterates $\{\dot{\theta}_b^t\}$ satisfy descent lemma, which is

$$\hat{\mathcal{L}}_b(\dot{\theta}_b^t) \leq \hat{\mathcal{L}}_b(\theta_b^{t-1}) - \frac{\eta}{2}\|\nabla\hat{\mathcal{L}}_b(\theta_b^{t-1})\|_2^2$$

Now, since $\hat{\mathcal{L}}_z$ is non-negative,

$$\hat{\mathcal{L}}_b(\theta_b^t) - \hat{\mathcal{L}}_z(\theta^*) \leq \frac{\rho_b}{\Delta_b^t}\hat{\mathcal{L}}_b(\theta_b^{t-1}) - \hat{\mathcal{L}}_z(\theta^*)$$

$$\leq \frac{\rho_b}{\Delta_b^t}\left(\hat{\mathcal{L}}_b(\theta_b^{t-1}) - \hat{\mathcal{L}}_z(\theta^*)\right) \quad \text{since } \rho_b/\Delta_b^t < 1$$

Using the above recurrence we get,

$$\hat{\mathcal{L}}_b(\theta_b^T) - \hat{\mathcal{L}}_z(\theta^*) \leq \left(\prod_{t=1}^{T} \frac{\rho_b}{\Delta_b^t}\right)\left(\hat{\mathcal{L}}_b(\theta_b^0) - \hat{\mathcal{L}}_z(\theta^*)\right)$$

Since $\delta_b' = \min_t \Delta_b^t, \forall t$, then for desired error $\epsilon$ on the upper bound we get,

$$T \geq \frac{1}{\log(\delta_b'/\rho_b)} \log\left(\frac{\hat{\mathcal{L}}_b(\theta_b^0) - \hat{\mathcal{L}}_z(\theta^*)}{\epsilon}\right)$$

$\square$

**Proposition D.5.** *(Never Trade, Linear Case) Let agents never trade and ignore the market i.e. $I_a^t = 0, I_b^t = 0$ for all $t$. Suppose they run individual gradient descent on functions $\hat{\mathcal{L}}_a, \hat{\mathcal{L}}_b$ using exact line search. Then for any $\epsilon \in (0,1)$ at the end of round $T$, we have $\hat{\mathcal{L}}_z(\theta_b^T) - \hat{\mathcal{L}}_z(\theta^*) \leq \epsilon$, for $T \geq \frac{1}{\log\left((\rho_b+1)/(\rho_b-1)\right)} \log\left(\frac{\hat{\mathcal{L}}_b(\theta_b^0) - \hat{\mathcal{L}}_z(\theta^*)}{\epsilon}\right)$.*

**Discussion.** First, we note that in both settings (always trade and never trade) the convergence rate is $\mathcal{O}(\log(\frac{1}{\epsilon}))$, which is expected since the loss function getting minimized is strongly-convex. The differences are mainly visible in the constant factors. Most importantly, the leading constant in the result of Theorem D.4 depends on $\delta'_b$— the minimum improvement ratio over $T$ runs. A higher improvement ratio implies a smaller $T$ (i.e. faster convergence) in the worst-case scenario. This justifies the benefit of purchasing the parameters in this setting. In contrast, the leading constant in the result of Proposition D.5 only depends on $\rho_b$. We further note that $\frac{\rho_b+1}{\rho_b-1} \geq \frac{1}{\rho_b}$, so to claim better rate in Theorem D.4 we would need $\delta'_b \geq \frac{\rho_b^2+\rho_b}{\rho_b-1}$.

# E  Experimental Details

In this section, we place more details about our experimental setups. All the implementation is open sourced at `https://github.com/SprocketLab/parameter-market`.

## E.1  Data Endowment

In the experiments for collaborative setting (Sec. 6.1), agents are limited to collecting a part of dataset on MNIST [33], CIFAR10 [34], and TinyImageNet [35]. For MNIST and CIFAR10, we make agent $A$ collect 10% of data points from class $0 \sim 4$, but obtain full data points from class $5 \sim 9$. For TinyImageNet, agent $A$ collects 10% of data points from class $0 \sim 99$, but obtains full data points from class $100 \sim 199$. Agent $B$ collects data points in the opposite way. We give broker full data points to evaluate agents' model performance (i.e. $\hat{\mathcal{L}}_z(\theta)$) and to inform gain-from-trade in the try-before-purchase mechanism.

## E.2  Model Training Settings

MLP models are trained with 5 hidden layers, each comprising 32 units, using ReLU activations between layers and no normalization. We employ SGD as the optimizer with a learning rate of $10^{-2}$. ResNet20 models are trained from scratch and the optimizer used is Adam with a learning rate of $10^{-2}$. In both settings, the batch size is set to 256.

# F  Trading Guidance in the Market

In the following paragraphs, we provide more experimental results. We use the collaborative setting to trade entire parameter sets and apply model alignment. We validate that (i) trading offers instantaneous improvements, (ii) trading after every training epoch results in the fastest convergence, (iii) trading makes agents who obtain fewer data earn more improvements relatively, and (iv) trading asynchronous parameters enhance performance as well. Last but not least, we generalize our setting to a multi-agent market and show parameter trading enables every agent in the market to achieve higher accuracy performance compared to out-of-market training.

**How does the epoch to start trading affect the convergence?**  We study this using two MLPs with MNIST datasets. Figure 6 shows the comparison of agents' testing loss when they begin trading after different epochs, namely { 10, 20, 30, 40 }. Our results indicate that joining the market at *any epoch* provides immediate improvements. Additionally, training parameters and then start trading at an earlier epoch results in faster convergence (i.e. starting to trade after epoch 10 leads to the highest accuracy performance at the end).

**How does the trading frequency affect the convergence?**  Next, we examine how the trading frequency affects convergence. Our experiment asks agents to trade after every { 1, 7, 10 } epoch and compares their testing loss to agents who don't trade at all. The results, shown in Figure 7, indicate that the optimal approach for agents is to trade after *every epoch*, which leads to the fastest convergence. Despite the fact that testing loss increases during periods when agents don't trade, they are still able to achieve lower losses overall.

**How does data endowment affect trading benefits?**  We also experiment with different levels of data endowment, ranging from 10% to 80%, and investigate the impact of data endowment on

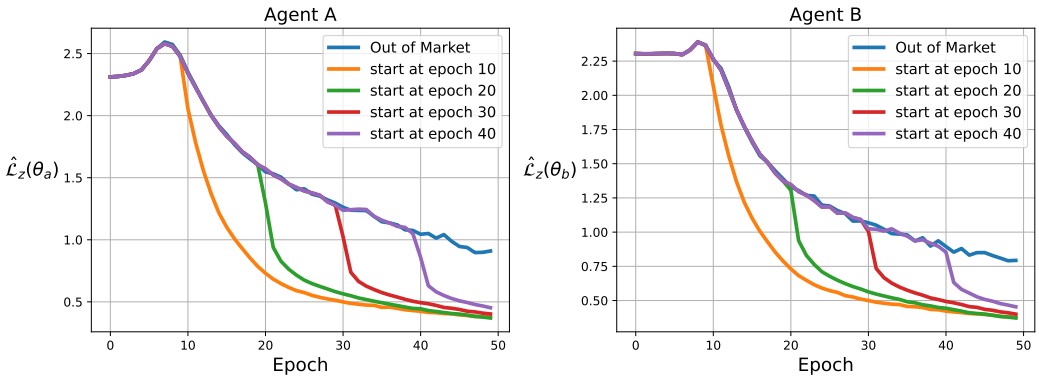

Figure 6: Joining the market at any epoch provides *instantaneous* trading benefits.

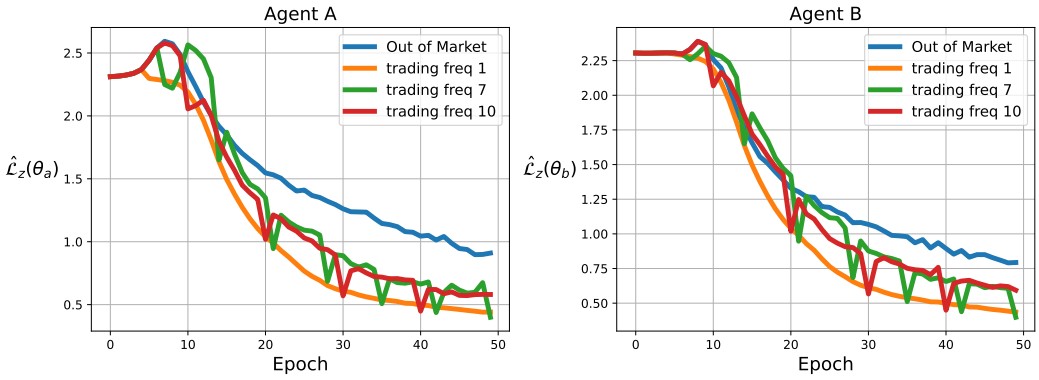

Figure 7: Trading parameters after *every epoch* gives the fastest convergence.

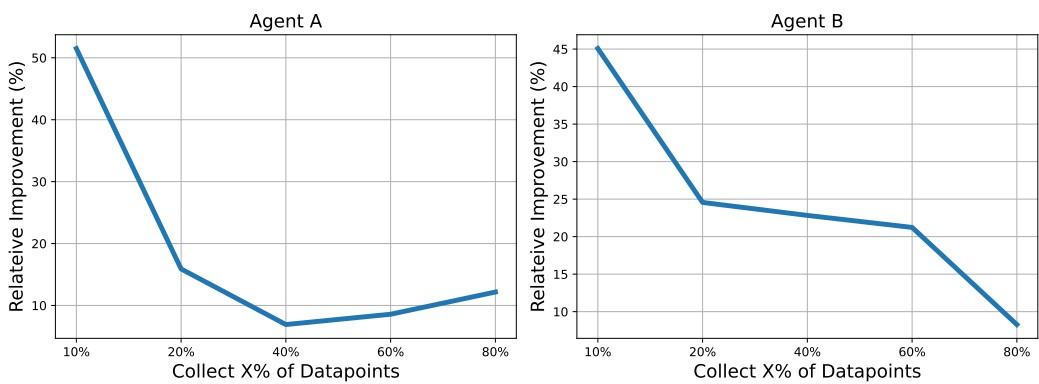

Figure 8: **More limited access to data earns more trading benefits**. Relative improvements are computed by the empirical testing loss compared to out-of-market agents.

trading benefits. Agents are set to collect these percentages of data points for half of the classes. The results, shown in Figure 8, indicate that agents who are more limited in access to data can benefit more from trading relatively. When agent $A$ and agent $B$ endowed with 10% data points can earn the most trading benefits through improving testing loss by **51.5%** and **45.1%**, respectively.

|  |  | Agent $A$ | Agent $B$ |
|---|---|---|---|
|  |  | Testing Acc. (%) | Testing Acc. (%) |
| MNIST + | out-of-market | 68.50% | 72.97% |
| MLP | w alignment | **88.78%** | **82.51%** |
| CIFAR10 + | out-of-market | 71.14% | 70.56% |
| ResNet20 | w alignment | **78.41%** | **73.40%** |

Table 3: We conduct a new experiment on asynchronous parameter trading. Both agents are asked to train the model for 60 epochs in total. Agent $A$ is instructed to delay trading. Results emphasizes the crucial role of brokers in aligning parameters and optimizing purchase weights to eliminate differences not only in synchronous but also in asynchronous parameter trading.

|  |  | Agent $A$ | Agent $B$ | Agent $C$ |
|---|---|---|---|---|
|  |  | Testing Acc. (%) | Testing Acc. (%) | Testing Acc. (%) |
| MNIST + | out-of-market | 68.07% | 73.79% | 76.38% |
| MLP | w alignment | **82.29%** | **77.08%** | **77.08%** |
| CIFAR10 + | out-of-market | 67.62% | 74.99% | 74.12% |
| ResNet20 | w alignment | **79.53%** | **77.77%** | **76.80%** |
| TinyImageNet + | out-of-market | 20.51% | 17.99% | 18.50% |
| ResNet20 | w alignment | **32.62%** | **32.07%** | **31.08%** |

Table 4: We generalize our setting to involve more agents. In the three-agent market, results also follow expectations. The proposed parameter trading is able to help agents achieve higher accuracy performance compared to conventional model training without trading (out-of-market).

**How does delayed parameter trading affect the performance?** Now we consider another practical scenario that permits asynchronous parameter trading. In this setting, both agents are asked to train the model for 60 epochs in total. Agent $B$ trains the model for 5 epochs, trades in the market for 50 epochs, and then trains the model for the remaining 5 epochs. On the other hand, agent $A$ is instructed to delay trading. Agent $A$ trains the model for 10 epochs and then trades in the market for 50 epochs. In Table 3, our results show that the parameter market allows delayed agent actions to enhance performance as well, with trading in alignment yielding optimal accuracy as expected. This emphasizes the crucial role of brokers in aligning parameters and optimizing purchase weights to eliminate differences *not only in synchronous but also in asynchronous parameter trading*. Note as well that in Table 1, agents train for 5 epochs and then trade synchronously for 55 epochs, which have additional 5 epochs for training then trading. This explains the degraded performance of agent $B$.

**How does more agents involved affect trading benefits?** For the sake of simplicity and to demonstrate the viability of parameter trading, we present a two-agent market in the main body, which is a commonly used economic model for studying many settings. Nevertheless, it is straightforward to expand the proposed trading framework to include multiple agents. The trading logic can be generalized to look for trades that enable agents to make the largest advancements. This means purchasing parameters that can bring the largest gain-from-trade ($\Delta_u^t$). To validate this, we generalize our setting to involve more agents. We implement a three-agent market by reusing the same data endowment for agent $A$ and agent $B$ and display performance results in Table 4. We make a third agent, agent $C$, collect 10% of data points from the class $3 \sim 7$ in MNIST and CIFAR10, and 10% of data points from class $50 \sim 149$ in TinyImageNet. In the three-agent market, results show that the generalization functions as expected: the proposed market is able to help agents achieve higher accuracy performance compared to conventional model training without trading.

## G  Discussion

In this section, we discuss limitations, (potential) concerns, the needs of broker's computational infrastructure, and the common prior that we assume in the valuation.

### G.1 Limitations

There are two primary limitations toward building viable parameter markets.

First, the reliability of the broker is crucial to the success of the market. The broker must report trade gains and losses without any bias to ensure the market remains efficient. Suppose that the broker acts adversarially, rather than being a neutral third party as intended. In this case, the broker could potentially use models (or their combination) from both parties, could misinform one party or another of their potential benefit and engage in front-running, etc. Hence, the requirement that some entity is willing to serve as an impartial broker is one of the limits of our work.

To achieve this, we can motivate third parties to play this role by earning transaction fee to facilitate trades, or by giving other types of incentives [38]. On the other hand, such a limitation may not be substantial. In fact, having a reliable broker is **essential for any trading market**, just as it is in real-life scenarios (for example, stockbrokers help perform huge numbers of trades, and large-scale escrow organizations help aid in billion-dollar transactions). This similar requirement is accepted and used in practice in other settings, such as (non-machine learning) markets [38].

An additional limitation is that while agents are allowed to train models using various network architectures for a range of downstream tasks, the parameter sets to be traded require a certain alignment. We believe this limitation is not too severe: it can be overcome, for example, by distilling knowledge onto the same space and dimension for merging [39].

### G.2 (Potential) Concerns

One potential concern is about model copyright. In fact, one of the motivations for proposing parameter marketplace is precisely *the fact that trading data is more challenging due to the state of digital property rights*. Parameter trading, on the other hand, does not suffer from the same limitations, at least in many present legal frameworks. That is, parameters are typically owned by organizations or individuals training their models. Because of this, voluntarily trading parameters does not require breaking any type of copyright law. Broker is the only additional party to access to these. Besides, we note that our market permits the trading of subsets of parameters (see Sec. 6.1.2) and transferred parameters only if a trade is made, thereby safeguarding a certain degree of model confidentiality.

Another potential concern is how to prevent out-of-bounds parameter usage (i.e., how to ensure a buyer does not publicly release the purchased parameters, sell them to another unauthorized party). There is a wide array of existing research on this question. Applicable tools enable users to check the purchased weights by creating transaction hashes [40] and watermarks [41].

### G.3 Broker's Infrastructure

It is essential to concern the accessibility of computational resources for the broker. In our framework, the broker has two main computational requirements for conducting a trade. Firstly, the broker needs to help agents validate the model's performance through inference, which is generally less intensive than training [42], and the amount of validation data needed does not need to be high [43]. Second, the broker assists in aligning parameters for both parties using an approximation algorithm for a bilinear assignment problem [44]. This algorithm has been shown to perform efficiently in experiments. Hence, we believe that the computational needs can be met by charging transaction fees to the agents. In other words, the computational workload for the broker is not a significant issue and can be managed.

### G.4 Common Priors in Valuation

The common priors when agents are pricing are that buyer values parameters and sets a price arising from gain-from-trade, where we write $v_u(\dot{\theta}_{u'}^t) = \Delta_u^t$, and this price is derived from a known probability distribution. Both are inspired by Myerson's auction work [23], designed to help sellers estimate the values of trades. The first prior is reasonable, since all agents consider trading benefits while assessing a parameter trade. The more they stand to gain, the higher the price they are willing to pay. Therefore, to maximize the seller's revenue, it is best to set the price as close as possible to the buyer's price. Then in Theorem 4.1, we help agents to bound the other's valuation. Second, the notion that the buyer's price is a random variable drawn from a probability distribution is standard and commonly used in economic modeling.