# OpenReview forum: "Train 'n Trade: Foundations of Parameter Markets"
_NeurIPS.cc/2023/Conference — NeurIPS 2023 poster_

### Official Review · Reviewer_Mu3U · 2023-07-05

**Soundness:** 4 excellent
**Presentation:** 4 excellent
**Contribution:** 4 excellent
**Rating:** 6
**Confidence:** 3

**Summary:**

This paper introduces a novel concept called *parameter markets*, which serves as a platform for exchanging parameters learned by machine learning models. In this framework, agents have the option to engage in parameter trading, to achieve (1) mutual benefits through collaboration or (2) monetary gains by simply selling these parameters. The key contributions of this paper involve the formulation and design of these parameter markets. This encompasses defining the value of parameters, establishing pricing mechanisms, and outlining how transactions can occur. The authors explore both competitive and collaborative scenarios and illustrate the advantages agents can obtain by opting in to trade through the marketplace.

**Strengths:**

1. The idea to trade parameters is novel (to my best knowledge) and is quite different from data markets. The key benefit from parameter markets is that agents can train models for their own tasks and can yet benefit from training runs of each other. This is a very relevant problem to study.
2. The formulation is thorough and the empirical experiments are adequate to demonstrate the claims made in the paper
3. The paper is well written and easy to follow.

**Weaknesses:**

1. This work is a bit existential - this framework works if *model privacy and alignment are assured*.
2. Theorem 4.1 relies on the broker having access to $\theta^*$ which is a strong assumption.
3. There is no discussion relating to incentives (i.e. agent misreporting their valuations to benefit from the trades).Perhaps this paper could benefit from adding a mechanism design angle to this framework.



**Questions:**

1. Refer to Weaknesses.
2. It's unclear why Cobb-Douglas utility function is used.

**Limitations:**

To my best knowledge, there isn't an immediate potential negative societal impact of their work. This paper, however, doesn't have an explicit weakness section (relating to their framework).

---

> ### Author Rebuttal · Authors · 2023-08-09
>
> ### Response to Reviewer Mu3U
> We are grateful for the review, the kind words, and the positive assessment. We address your questions and concerns below.
> * **On assurance of model privacy and alignment.**
>     * This is true! We do not view this as a limitation, however. Vast research resources are being expended in assuring privacy [1, 2] and alignment can be performed [3, 4, 5, 6]. The motivation of our work is to discover, _if_ these concerns can be resolved, whether parameter marketplaces are viable at all. We believe that an answer to this question, whether positive or negative, is extremely valuable.
>     * We note, in addition, that the progress required in alignment and privacy may not be out of reach. According to the findings presented in Sec. 5.1, while trading with model alignment yields optimal outcomes, basic convex interpolation is also sufficient in achieving quicker convergence and enhancements. Furthermore, in our trading system, parameters are secured and only exchanged if both parties agree on the agreed-upon price, thus ensuring a type of privacy in the market.
> * **On accessing true parameters.**
>     * We agree! Having knowledge of the true parameter $\theta^*$ to reveal the gain-from-trade $\Delta^t_u = ||\dot\theta^t_u - \theta^*||^2_2 / ||\bar\theta^t_u - \theta^*||^2_2$ is a strong assumption. However, the result using it is only a specific example that can assist sellers in assessing the transaction and providing theoretical insights (Theorem 4.2). In practice, **there are a variety of ways to determine the value of a trade** based on factors such as the agent's preference, resource budget, and relative performance improvement. Furthermore, for more practical purposes, we present a generalized convergence analysis in Appendix D.2, where the gain-from-trade is calculated by subtracting the empirical loss before and after a trade, $\Delta^t_u = \hat{L}(\dot\theta^t_u) - \hat{L}(\bar\theta^t_u)$, removing the need for the broker to possess knowledge of the true parameter $\theta^*$.
> * **On agent’s incentives and honesty.**
>     * Thank you! We note another finding in Sec. 5.2, where the seller estimates the value solely using the lower bound they found according to Theorem 4.1. The resulting negotiated price creates a discrepancy with the price in the optimal scenario where both parties report their valuations truthfully without any estimation, **highlighting the significance of revealing accurate parameter values and justifying the need for incentives.**
> * **On the perspective of mechanism design.**
>     * Thank you for your suggestion! This is an interesting approach to strengthen our work, and we will include this in our updated draft.
> * **On the clarity of Cobb-Douglas utility function.**
>     * The concept behind price negotiation is to split a fixed joint surplus ($U_a \times U_b$) while maximizing the revenue for both parties equally (as the power of $U_a$ and $U_b$ is the same). The Cobb-Douglas utility function is ideal for this purpose. In addition, it has several convenient mathematical properties. For instance, it is strictly quasiconcave, which means it has a unique maximizer and an optimal solution. Moreover, it demonstrates the diminishing marginal rate of substitution as the negotiated price changes. Lastly, when expressed in log form, it serves as the first Taylor approximation of a convex production function, thereby eliminating the need for extensive knowledge of the underlying production function.
>
> [1] Zhu et al, "Privacy-preserving Decentralized Federated Deep Learning", ACM TURC 2021. \
> [2] Pasquini et al, "On the (In)security of Peer-to-Peer Decentralized Machine Learning", IEEE Symposium on Security and Privacy (SP), 2023. \
> [3] Stoica et al, "ZipIt! Merging Models from Different Tasks without Training", arXiv:2305.03053. \
> [4] Wortsman et al, "Model soups: averaging weights of multiple fine-tuned models improves accuracy without increasing inference time", ICML 2022. \
> [5] Singh and Jaggi, "Model Fusion via Optimal Transport", NeurIPS 2022. \
> [6] Qin et al, "Exploring Mode Connectivity for Pre-trained Language Models", EMNLP 2022.

---

> > ### Comment · Reviewer_Mu3U · 2023-08-16
> >
> > Thanks for the response!

---

> > > ### Author Response · Authors · 2023-08-19
> > >
> > > Thank you for engaging with our rebuttal! We are excited to incorporate the finding regarding agent's incentives and clarity of utility function into the paper. We are eager to provide any further clarifications or address any additional questions. Thank you!

---

### Official Review · Reviewer_Tv8J · 2023-07-06

**Soundness:** 2 fair
**Presentation:** 2 fair
**Contribution:** 2 fair
**Rating:** 5
**Confidence:** 3

**Summary:**

* This paper proposes an economic framework for trading parameters of prediction models.
* Interaction is modeled as a brokered marketplace - Each agent $u$ trains a model characterized by parameters $\theta_u$ using gradient descent. At each time-step $t$:
  * Each agent performs a gradient descent step on the previous parameters $\theta_u^{t-1}$, to obtain $\dot{\theta}_u^t$.
  * After the GD step, agents relay their model parameters $\dot{\theta}_u^t$ to a  trusted broker, which calculates the optimal linear interpolation between them, yielding the possibly-improved combination $\bar{\theta}_u$ for each agent. The broker indicates for each agent the gain-from-trade $\Delta_u^t$.
  * The agents participate in Nash bargaining to decide on the value exchange, set trading prices, and $\theta^t_u$ is decided. Each agent has a valuation function $v_u$ for their own parameters, and the other agent’s parameters. Utility function is assumed to be fully known by the broker.
* Analysis:
  * Analysis is restricted to the two-agent setting $u\in\{A,B\}$, and a specific form for gain-from-trade $\Delta_u^t$ is assumed. It is assumed that each seller has complete knowledge of the valuation prior, such that Myerson’s revenue-maximizing pricing mechanism (virtual valuations) can be applied based on the information relayed from the broker ($\alpha$, $\beta$, $\Delta_a^t$ - By Theorem 4.1).
  * It is assumed that the broker has complete knowledge of the optimal model parameters in advance ($\theta^*$). Combining with the assumptions about the agents’ utility function, a closed-form expression is obtained for the monetary transfer after Nash bargaining.
  * In Section 4.1, two upper bounds on convergence rates are presented - One for a scenario where agent $B$ always trades (Theorem 4.2), and one for a scenario where both agents don’t trade (Proposition 4.3). The “always-trade” bound (Thm 4.2) is lower, and the authors claim that this explains why trading is sometimes beneficial.
* Two sets of experiments are presented:
  * The first set of experiments is an evaluation of the data-sharing setting, without taking parameters into account (Section 5.1.1 - complete networks, Section 5.1.2 - subsets of networks), on real-world vision datasets (MNIST, CIFAR10, TinyImageNet).
  * Finally, the value of trading is evaluated using a synthetic model, and positive results are reported for both cooperative and competitive scenarios.


**Strengths:**

* Problem is well-motivated.
* Leverages recent deep insights about the structure of contemporary learning problems.
* The breadth of work is substantial, and it contains a number of new definitions, theoretical analyses, and empirical evaluations.
* Code is provided.


**Weaknesses:**

* Some key assumptions are not well supported:
  * Valuation functions $v(\theta)$ are assumed to be fully known by the broker. For contemporary neural nets, the dimensionality of $\theta$ could be in the order of billions (or even trillions, such as in GPT4), and therefore the parameter valuation would be a function $v:\mathbb{R}^{10^9}\to\mathbb{R}$ - Which may be prohibitively expensive to represent, store and compute.
  * Broker is assumed to know the true model parameters $\theta^*$ in advance, which in practice may render the whole learning process unnecessary. In L214, it is claimed that knowing $\theta^*$ “is not necessary in practice”, but I did not find further support of this claim.
  * Very specific functional forms are assumed for gain-from-trade but are not sufficiently justified.
  * A common prior on agent valuation is assumed (L208), however it is not discussed whether having such prior is realistic and how it would be implemented.
* Mathematical soundness concern regarding one of the claims: In section 4.1, authors claim that trading is beneficial by comparing two performance upper bounds (Theorem 4.2, Proposition 4.3). However, the difference between two upper bounds does not indicate the relation between the true quantities unless they are tight (in other words, proving that $a<100$ and $b<50$ does not indicate that $a>b$ unless there are matching lower bounds for $a$ and $b$). Therefore, if this observation is true, the analysis in Section 4.1 does not address the posed question (whether participation leads to better convergence).
* First set of experiments (Sec 5.1.1, 5.12) only seem to demonstrate the ability to interpolate model parameters, but do contain simulation of parameter trading. Economic behavior is only demonstrated on the synthetic dataset. As the benefit of parameter interpolation has already been demonstrated in the literature to some extent, I did not understand the contribution of the results in sections 5.1.1, 5.1.2 to the understanding of parameter-trading markets.
* Paper only analyzes the case of two agents $\{A,B\}$. Unclear how results extend to multiple agents.
* Trading parameters during the gradient descent process would require agents to train models simultaneously, or suffer significant delays due to trading. This may make such trading impractical.


**Questions:**

* How would the model extend to more than 2 agents? Which results can be used as-is, and which need to be generalized? Would the system behave qualitatively differently in any way when the number of agents is increased?
* What are the computational requirements from the broker?
* Is it possible to quantify the economic implications of the broker not being able to interpolate successfully?


**Limitations:**

I feel that limitations could be discussed at more depth. The key assumptions are added gradually throughout the paper, and it is hard to keep track. I feel that the paper can greatly benefit from a thorough discussion of limitations.

---

> ### Author Rebuttal · Authors · 2023-08-09
>
> ### Response to Reviewer Tv8J
> Thank you for your clear summary and thoughtful review! We appreciate the kind words. We answer your questions below and include two new experimental results: multi-agent market and asynchronous parameter trading.
> * **On valuation function.**
>     * Thanks for pointing this out. In fact, the valuation function is not exposed to anyone in the market (see Ln 173 - Ln 175), but the valuation of trade (a scalar) will be disclosed to the broker for negotiation purposes. Additionally, the valuation function is quantifying the value of parameter sets based on gain-from-trade, which is written in the form of losses, for example in Ln 161. Hence, agents do not need to plug in high-dimensional parameter sets and compute the value. They value a potential trade by knowing the gain-from-trade.
> * **On knowing the true parameter.**
>     * However, it is only for the sake of offering a concrete example of assisting sellers in assessing the transaction and providing theoretical insights (Theorem 4.2). In practice, **there are a variety of ways to determine the value of a trade** based on factors such as preference, budget, and relative returns. For more practical purposes, we present a generalized convergence analysis in Appendix D.2, where we remove the need for the broker to possess knowledge of $\theta^*$.
> * **On the form of gain-from-trade.**
>     * The gain-from-trade in our case study can be seen as the relative improvement if a trade is made. It should be noted that the gain-from-trade can take different forms and **is not limited to a specific definition**. For instance, it can be defined as the difference between an agent's loss before and after trading (as seen in Ln 161). Excitingly, **any method that quantifies the benefit of trading before and after** can be generalized into our pricing mechanism.
> * **On common prior.**
>     * The common prior (Ln 205): buyer sets a price arising from gain-from-trade and this price is derived from a probability distribution. **Both are inspired by Myerson's auction work, designed to help sellers estimate the values of trades**. The first prior is reasonable, since all agents consider the gain-from-trade while assessing a trade. Second, the notion that buyer's price is drawn from a probability distribution is standard in economic modeling [1]. We clarify this in our updated draft.
> * **On the upper bound of convergence rate.**
>     * We appreciate the comment. We have reviewed and revised our draft accordingly. While the upper bound on the end round $T$ may not necessarily lead to faster convergence, participating in the market provides agents with sufficient incentive and the assurance of achieving a smaller $T$ in the worst-case scenario. The discrepancy arises from the presence of gain-from-trade. We note that this is not unusual: comparing the upper bounds of the convergence rates (worst cases) is standard in the optimization.
> * **On the contribution of Sec. 5.1.1 and Sec. 5.1.2.**
>     * To the best of our knowledge, previous research has mainly focused on merging entire trained models at the end of the training process [2, 3], but there has been limited investigation into aligning parameters or subsets of parameters during training. This is a crucial factor in establishing a functional marketplace, and it is worthwhile to investigate such potential benefits. Hence, we demonstrate the feasibility of parameter trading while training in Sec. 5.1.1 and 5.1.2.
> * **On extension to multiple agents.**
>     * For the sake of simplicity and demonstrate the viability, we presented a two-agent market, which is a commonly used economic model for studying many settings. Nevertheless, it is straightforward to include multiple agents. The trading logic can be generalized to look for trades that enable agents to make the largest advancements. To validate this, **we implemented a three-agent market and present detailed experimental setups** in our general response. Results show that the generalization functions as expected.
> * **On asynchronous trading.**
>     * Thank you for bringing up this important point. Indeed, this area has been extensively studied in the distributed training literature [4]. The typical solution is to permit asynchrony. **We conducted asynchronous parameter trading and offer detailed experimental setups** in the general response. Results show that the parameter market allows delayed agent actions to enhance performance as well. Since the presence of the broker helps agents align parameters and optimize purchase weights, **the potential issues caused by asynchronous training can be mitigated**.
> * **On computational requirements for broker.**
>     * The broker has two main computational needs. Firstly, the broker needs to validate model's performance through inference, which is generally less intensive than training. Second, the broker assists in aligning parameters for both parties using an approximation algorithm, which has been shown to perform efficiently in experiments. In other words, the **computational workload for the broker is not a significant issue and can be managed**.
> * **On quantifying interpolation failure.**
>     * This is a great question, as the weights purchased may not be useful for agents if they are built for very unrelated tasks, which can result in a negative gain-from-trade and lead to agents stopping their purchases, ultimately reducing market efficiency. To measure such failures, one can establish prior knowledge of task interdependence. As mentioned in Section 5.1.4, the advantages of trading diminish when the tasks are not related.
>
> [1] Chawla et al, "Algorithmic Pricing via Virtual Valuations", EC 2007. \
> [2] Stoica et al, "ZipIt! Merging Models from Different Tasks without Training", arXiv:2305.03053. \
> [3] Qin et al, "Exploring Mode Connectivity for Pre-trained Language Models", EMNLP 2022. \
> [4] Wang et al, "Asynchronous Training Schemes in Distributed Learning with Time Delay", arXiv:2208.13154.

---

> > ### Comment · Reviewer_Tv8J · 2023-08-18
> >
> > Thank you for the detailed response. I appreciate the discussion, and the additional empirical results. I’m increasing my rating to 5.

---

> > > ### Author Response · Authors · 2023-08-19
> > >
> > > Thank you for increasing your score and engaging with our rebuttal. We are excited to include the new experimental results in the updated version of our paper. We would love to answer any additional questions! Thank you!

---

### Official Review · Reviewer_iN1r · 2023-07-06

**Soundness:** 2 fair
**Presentation:** 2 fair
**Contribution:** 2 fair
**Rating:** 4
**Confidence:** 2

**Summary:**

The paper investigates how to design a marketplace for model parameters. The marketplace consists of agents training models for potentially different objectives and a trusted third-party broker. The broker receives the model parameters from each agent, assesses (and informs each agent of) the loss achieved by the merged model, and helps determine prices using Nash bargaining.

The paper provides a theoretical and empirical analysis of this framework. In particular, they prove a bound on the gain from trade for linear models and they analyze the effectiveness of buying parameters in terms of improving training convergence.  They empirically assess their framework on an image classification task. They validate that the merging method of the broker (Ainsworth et al., 2022) improves loss, and they evaluate the effectiveness of their pricing strategy.


**Strengths:**

The idea of trading subsets of parameters in a marketplace seems novel and interesting. Parameter marketplaces seem to contrast with classical approaches of data marketplaces, where agents buy data from each other, or model marketplaces, where agents directly purchase models or model access.

**Weaknesses:**

The proposed framework seems to rely on there being a trusted broker facilitating the trades and assessing and reporting the gains of merging parameters to the agents. In reality, this seems to be a strong assumption, as it is not clear if (a) a broker can be trusted with all of the parameters, and (b) if a broker will be able to assess the gain of merging models without significant data and infrastructure. See question below.

The technical contribution of the paper seems limited. The paper combines existing work on merging and aligning models (e.g. Ainsworth et al., 2022) and classical approaches for Nash bargaining. However, the paper seems to use these approaches out-of-box and does not seem to present significant new technical contributions.

The paper is a bit difficult to follow and the exposition could be improved. For example, the results in Section 4 use a lot of notation and the qualitative insights are a bit difficult to infer.


**Questions:**

How might the framework be modified to avoid the reliance on a trusted broker, e.g., via a more decentralized approach?


**Limitations:**

The authors adequately addressed the limitations.

---

> ### Author Rebuttal · Authors · 2023-08-09
>
> ### Response to Reviewer iN1r
> We are grateful for your review and for describing our framework as novel and interesting. We address your questions in the response below and have updated our paper!
> * **On trusting brokers.**
>     * Indeed, having a reliable broker is **essential for any trading market**, just as it is in real-life scenarios (for example, stockbrokers help perform huge numbers of trades, and large-scale escrow organizations help aid in billion-dollar transactions). One area of interest is how to motivate third parties to play this role, a question that is frequently studied by economists and business researchers [1]. Examples of this are research into how to set fees for brokers, what other types of incentives can be found, etc.
>     * We view this question as orthogonal to this work, and instead simply presume that such brokers exist. This supposition is commonly employed and has been substantiated by a number of prior research papers on trading markets [2, 3, 4]. We note, in addition, that our market permits the trading of subsets of parameters and transferred parameters only if a trade is made, thereby safeguarding a certain degree of model confidentiality.
> * **On broker infrastructure.**
>     * This is a great question! We agree that it is essential to concern the accessibility of computational resources for the broker, but we do believe this is not a major challenge. The reason for this is that **brokers can be paid from brokerage fees** to a sufficient extent to meet their computational needs. Note that these requirements need not be prohibitive: our trading framework does not ask the broker to train models, only to validate model performance through inference, which typically requires fewer resources [5]. Additionally, the amount of validation data needed does not need to be high [6]. Moreover, in general, the alignment approach we employ is computationally efficient. As a result, the computational requirements for the broker are manageable.
> * **On contributions.**
>     * Our focus is on developing a novel trading framework that contributes to creating parameter markets and demonstrating the feasibility of trading sets of parameters. The techniques used in this framework are flexible and can include more alignment approaches [7, 8, 9, 10] and be generalized to any other utility functions for price negotiation [11]. Our findings are important: they suggest that **the entire approach of parameter trading is viable**. Indeed, this is the major contribution of our work.
>     * In addition, through empirical evidence, we have found that trading parameter sets can also lead to faster convergence and improved model performance (refer to Sec. 5.1.2)---a further contribution. Finally, as far as we are aware, this is the first work that successfully monetizes parameter sets and treats them as commodities to be traded. Such a market can be used in the current large language model community: by engaging in parameter trading, individuals or start-ups are able to purchase pre-trained parameter sets instead of building from scratch.
> * **On clarifying results and takeaways.**
>     * Sec. 4 offers concrete examples of market instances. We highlight two main results and have clarified this further in our draft.
>         * **How does a seller value a trade?** We study the case of linear models and help the seller provide a valuation to the broker by using virtual valuation. Theorem 4.1 reveals that the seller can estimate their own valuation by bounding the buyer's valuation based on information disclosed by the broker. We empirically validate the success of price estimation procedure to monetize parameters in Sec. 5.2.
>         * **How effective is buying parameters from a theoretical perspective?** To investigate the effectiveness of parameter trading, we offer a convergence analysis. Theorem 4.2 states that more gain-from-trade in the market results in faster convergence, which supports the trading incentives to participate in the market. Additionally, we offer a generalized form of convergence analysis with L-smooth functions in Appendix D.2, which also demonstrates similar results that more gain-from-trade leads to quicker convergence. These results can be justified by empirical evidence in Sec. 5.1.
> * **On avoiding a trusted broker through decentralization.**
>     * We agree! A decentralized approach can indeed eliminate the need for a centralized broker. However, there are downsides to this approach as well. We would require new ways to motivate decentralized agents to perform the work. This may cut against our goal of minimizing expenses for individuals. Nevertheless, we think this is an interesting question and consider it for further investigation.
>
> [1] Yavas, "Economics of Brokerage: An Overview", Journal of Real Estate Literature, Vol. 2, No. 2, 1994. \
> [2] Liu et al, "Dealer: An End-to-End Model Marketplace with Differential Privacy", Proceedings of the VLDB Endowment, Vol. 14, No. 6, 2021. \
> [3] Azcoitia and Laoutaris, "Try Before You Buy: a Practical Data Purchasing Algorithm for Real-World Data Marketplaces", DE 2022. \
> [4] Chen et al, "Towards Model-based Pricing for Machine Learning in a Data Marketplace", SIGMOD 2019. \
> [5] Aminabadi et al, "DeepSpeed- Inference: Enabling Efficient Inference of Transformer Models at Unprecedented Scale", SC 2022. \
> [6] Kossen et al, "Active Testing: Sample–Efficient Model Evaluation", ICML 2021. \
> [7] Stoica et al, "ZipIt! Merging Models from Different Tasks without Training", arXiv:2305.03053. \
> [8] Wortsman et al, "Model soups: averaging weights of multiple fine-tuned models improves accuracy without increasing inference time", ICML 2022. \
> [9] Singh and Jaggi, "Model Fusion via Optimal Transport", NeurIPS 2022. \
> [10] Qin et al, "Exploring Mode Connectivity for Pre-trained Language Models", EMNLP 2022. \
> [11] McFadden, "Constant Elasticity of Substitution Production Functions", The Review of Economic Studies, Vol. 30, No. 2, 1963.

---

> > ### Comment · Reviewer_iN1r · 2023-08-16
> >
> > Thanks to the authors for responding to my questions. I appreciate the response and its discussion of the trusted broker assumption and the broker infrastructure. I have raised my score from a 3 (reject) to a 4 (borderline reject).

---

> > > ### Author Response · Authors · 2023-08-19
> > >
> > > Thank you for raising your rating---we appreciate your engagement with our rebuttal! We have included new experimental results and added a discussion section on the study of brokers, trust, incentives, and infrastructure in our updated draft. We would like to ask whether there are any further questions we can answer. Thanks you!

---

### Official Review · Reviewer_91tn · 2023-07-10

**Soundness:** 3 good
**Presentation:** 3 good
**Contribution:** 3 good
**Rating:** 6
**Confidence:** 3

**Summary:**

The paper proposes a framework for collaborative and competitive parameter trading among deep learning agents. The authors conduct experiments to validate the effectiveness of the proposed framework in improving the performance of the agents. The experiments show that even when the agents are training on different tasks, they can still benefit from trading parameters. The authors also validate the proposed pricing mechanism in a competitive scenario. The paper makes contributions in demonstrating the potential of parameter trading to improve the performance of deep learning agents and providing a framework for collaborative and competitive trading.


**Strengths:**

- The paper proposes a novel framework for collaborative and competitive parameter trading
- The experimental results are promising.
- The experiments show that even when the agents are training on different tasks, they can still benefit from trading parameters.

**Weaknesses:**

- The authors do not provide an analysis of the limitations of the proposed framework, which could help identify potential issues in real-world scenarios.
- The experiments could benefit from a more extensive evaluation of the proposed framework's performance by comparing it with more baselines.
- The paper could benefit from a more detailed explanation of the proposed pricing mechanism and its implementation.
- The authors do not discuss the potential ethical implications of parameter trading, which could be relevant in real-world applications.

**Questions:**

- Why is parameter trading necessary?
- How to resolve the legal concerns on copyrights?


**Limitations:**

More limitations should be discussed.

---

> ### Author Rebuttal · Authors · 2023-08-09
>
> ### Response to Reviewer 91tn
> Thank you for finding our framework novel and the results to be encouraging, particularly in terms of trading parameters with different purposes. We appreciate your thoughtful review!
> * **On limitations.**
>     * There are two primary limitations toward building viable parameter markets. We discuss these in the paper and have included additional clarifications. First, the reliability of the broker is crucial to the success of the market. The broker must report trade gains/losses without any bias to ensure the market remains efficient. One limits of our work is that some entity is willing to serve as a broker with these properties.
>     * On the other hand, such a limitation may not be substantial. In fact, a similar requirement is accepted and used in practice in other settings, such as (non-machine learning) markets [1].
>     * An additional limitation is that while agents are allowed to train models using various model architectures for a range of downstream tasks, the parameter sets to be traded require a certain alignment. We believe this limitation is not too severe: it can be overcome, for example, by projecting the parameters onto the same space for merging [2].
> * **On a new baseline.**
>     * Thank you for your suggestion. We have conducted another baseline test (FedAvg) and describe the setup in the general response. Results are presented in Table 1 in our attachment, and they confirm the significance of having a trusted broker in parameter trading. Without an intermediary broker to facilitate the trade, the performance of purchased weights is negatively impacted, as evidenced by the CIFAR10 + ResNet20 results.
> * **On further explanation of the pricing mechanism.**
>     * The discussion of pricing mechanisms is in Sec. 3.4, along with Sec. 4, where we provided a practical example to instantiate the market. We include additional explanations here and have updated our draft.
>     * The steps are as follows. Once agents become aware of the gain-from-trade information disclosed by the broker, they assess the value of parameters and negotiate prices accordingly. To negotiate prices, the broker solves a type of Nash bargaining problem. There are several ways to assess parameters, such as according to the agent's preference, training budget, and relative improvements. To provide a concrete study of how to value trades for the seller, we study trading in linear models and introduce virtual valuation for sellers. Finally, buyer's valuation can be bounded, enabling the seller to provide an estimated asking price for negotiation purposes.
> * **On potential ethical implications.**
>     * Under the conditions we require for our framework, we do not foresee ethical concerns around this work. However, if some of the assumptions we make are violated, we could envision certain ethical implications. For example, suppose that the broker acts adversarially, rather than being a neutral third party as intended. In this case, the broker could potentially use models from both parties, could misinform one party of their potential benefit and engage in front-running, etc. We have added a discussion of these possibilities to our updated draft.
> * **On why parameter trading is necessary.**
>     * Our motivation follows from the fact that the costs of building high-quality ML models have become prohibitively high, leaving only a few well-capitalized organizations in a position to build such models. Indeed, we see this in the space of large pre-trained models, where all other individuals can at best fine-tune or adapt these base models. **Our goal is to take a step towards reducing such costs**. The most intuitive approach is to perform distributed training. However, this would require fully open-sourced models, which many users may not be able to adhere to.
>     * How can we reduce costs, even when models aren't open-sourced and organizations may even be competing with each other? **Our solution is a type of parameter trading that can benefit all parties**. Buyers can purchase pre-trained parameter sets instead of building from scratch, reducing training expenses and leveraging the expertise of others. Sellers can profit from selling their parameters as a secondary source of income. The most exciting aspect of this work demonstrates that this simple solution works: we prove the effectiveness of parameter trading and allow buyers having faster convergence of losses. Sec. 5.2 demonstrates the success of our pricing mechanism and the opportunities for sellers to earn a profit.
> * **On legal concerns with copyright.**
>     * This is a great question! In fact, one of our motivations is precisely _the fact that trading data is challenging due to the state of digital property rights_. **Parameter trading, on the other hand, does not suffer from the same limitations**, at least in many present legal frameworks. That is, parameters are typically owned by organizations or individuals training their models. Because of this, voluntarily trading parameters does not require breaking any type of copyright law. Broker is the only additional party to access to these.
>     * One potential additional concern is how to prevent out-of-bounds parameter usage (i.e., how to ensure a buyer does not publicly release the purchased parameters, sell them to another unauthorized party). There is a wide array of existing research on this question. Applicable tools enable users to check the purchased weights by creating transaction hashes [3] and watermarks [4]. We discuss these in our updated draft.
>
> [1] Yavas, "Economics of Brokerage: An Overview", Journal of Real Estate Literature, Vol. 2, No. 2, 1994. \
> [2] Khanuja et al, "MergeDistill: Merging Language Models using Pre-trained Distillation", ACL-IJCNLP, 2021. \
> [3] Lawrenz et al, "Blockchain Technology as an Approach for Data Marketplaces", ICBCT 2019. \
> [4] Boenisch, "A Systematic Review on Model Watermarking for Neural Networks", arXiv:2009.12153.

---

> > ### Author Response · Authors · 2023-08-19
> >
> > Thank you for your thoughtful review and suggestions regarding framework limitations and baselines. Our response discusses limitations and we have included new baselines. We are eager to provide any further clarifications or address any additional questions.

---

### Author Rebuttal · Authors · 2023-08-09

### General Response

We are grateful for all the comments and constructive feedback on our work. Reviewers consistently commented that our proposed trading framework is **novel and well-motivated**. Reviewers 91tn, Tv8J, and Mu3U note that our experimental results as **promising and substantial**. In our revision, we adopted several reviewer-suggested clarifications and performed additional experiments, leading to a much stronger draft.

Additional experiments include:
* **A new baseline** [Reviewer 91tn]: we provide another baseline test (FedAvg [1]), which assumes that there is no broker involved in a trade to help agents align parameters and optimize their purchased weights. In the FedAvg method, the interpolated weight is determined by the portion of data assets an agent is endowed with, which is 0.5 in the case of our two-agent trading example. The results are presented in Table 1 in our one-page response. Results confirm **the significance of having a trusted broker** in parameter trading. Without an intermediary broker to facilitate the trade, the performance of purchased weights can be negatively impacted, as evidenced by the CIFAR10 + ResNet20 results.
* **Multi-agent markets** [Reviewer Tv8J]: we generalize our setting to involve more agents, where the broker helps agents seek for parameters that can bring the largest gain-from-trade to purchase. We implement a three-agent market by reusing the same data endowment for Agent $A$ and Agent $B$ and display performance results in Table 2. We make Agent $C$ collect 10% of data points from class 3 ~ 7 in MNIST and CIFAR10, and 10% of data points from class 50 ~ 149. In the three-agent market, results also follow expectations. The proposed parameter trading is able to help agents achieve higher accuracy performance compared to conventional model training without trading (out-of-market).
* **Asynchronous parameter trading** [Reviewer Tv8J]: we conduct a new experiment on asynchronous parameter trading. Both agents are asked to train the model for 60 epochs in total. Agent $B$ trains the model for 5 epochs, trades in the market for 50 epochs, and then trains the model for the remaining 5 epochs. On the other hand, Agent $A$ is instructed to delay trading. Agent $A$ trains the model for 10 epochs and then trades in the market for 50 epochs. In Table 3, our results demonstrate that the parameter market allows both agents to enhance their performance, with trading in alignment yielding optimal accuracy as expected. This emphasizes the crucial role of brokers in aligning parameters and optimizing purchase weights to eliminate differences **not only in synchronous but also in asynchronous parameter trading**. Note as well that in the paper, agents train for 5 epochs and then trade synchronously for 55 epochs, which have additional 5 epochs for training and trading. This explains the degraded performance of Agent $B$.

We have addressed reviewers' concerns and placed our comments in their respective threads below. Thank you for your questions and thoughtful reviews!

[1] McMahan et al, "Communication-Efficient Learning of Deep Networks from Decentralized Data", AISTATS 2017.

---

### Author Response · Authors · 2023-08-11
**Thank all reviewers again and keep further discussion**

Dear Area Chair,

We would love it if you could remind reviewers to kick off conversations with us!

We believe we could easily answer any remaining questions and make any further clarifications. Additional feedback from reviewers would be invaluable.

We would particularly love to engage with reviewers iN1r and Tv8J during the discussion period.
Thank you so much for your help!

The Authors

---

### Decision · Program_Chairs · 2023-09-21

**Decision:**

Accept (poster)

**Comment:**

This paper makes use of recent advances in model alignment to advocate for the idea of parameter markets, in which learned parameters can be traded by multiple entities. Reviewers found this prospect to be novel and intriguing, offering a viable alternative to existing markets for data or for entire models.

Reviewers have raised several concerns, the foremost being that technical novelty is low. Nonetheless, they acknowledge that the paper combines existing ideas in a useful way and applies them to a new problem domain. A second concern is that the proposed framework relies on strong assumptions, which some viewed as unrealistic - especially regarding the role and knowledge of the broker, but also on trust and infrastructure requirements. The authors' response was somewhat helpful in mitigating these, but reviewers still feel that the limitations and implications of the assumptions were not sufficiently discussed in the paper, with one reviewer feeling that the actual market mechanism was not entirely in line with how it was presented. More generally, the discussion regarding the paper's central "what if" assumptions - privacy, alignment, and idealized broker knowledge - requires more careful treatment than what is currently provided.

The authors are therefore highly encouraged to fix these issues, and to discuss not only the benefits of their approach, but also what can go *wrong* in such markets.

However, overall reviewers believe that the novelty of the idea of a market for parameters and the interest it promotes, coupled with the broad treatment and promising empirical results, are sufficient to merit acceptance. Reviewers were also appreciative of the new results presented in the discussion and in response to their questions. Hence, despite the above drawbacks, we believe this work has potential to spark interest in the community, and can serve to broaden discussion around economic aspects of machine learning.